# BioBO: Biology-informed Bayesian Optimization for Perturbation Design

**Yanke Li**[1,2*], **Tianyu Cui**[1*], **Tommaso Mansi**[1], **Mangal Prakash**[1†], **Rui Liao**[1†]
[1]Johnson & Johnson Innovative Medicine, [2]ETH Zurich

## Abstract

Efficient design of genomic perturbation experiments is crucial for accelerating drug discovery and therapeutic target identification, yet exhaustive perturbation of the human genome remains infeasible due to the vast search space of potential genetic interactions and experimental constraints. Bayesian optimization (BO) has emerged as a powerful framework for selecting informative interventions, but existing approaches often fail to exploit domain-specific biological prior knowledge. We propose Biology-Informed Bayesian Optimization (BioBO), a method that integrates Bayesian optimization with multimodal gene embeddings and enrichment analysis, a widely used tool for gene prioritization in biology, to enhance surrogate modeling and acquisition strategies. BioBO combines biologically grounded priors with acquisition functions in a principled framework, which biases the search toward promising genes while maintaining the ability to explore uncertain regions. Through experiments on established public benchmarks and datasets, we demonstrate that BioBO improves labeling efficiency by 25-40%, and consistently outperforms conventional BO by identifying top-performing perturbations more effectively. Moreover, by incorporating enrichment analysis, BioBO yields pathway-level explanations for selected perturbations, offering mechanistic interpretability that links designs to biologically coherent regulatory circuits.

## 1 Introduction

In vitro cellular experimentation with genomic interventions is a critical step in early-stage drug discovery and target prioritization. By perturbing genes and observing cellular responses, researchers can infer gene function and identify potential therapeutic targets (Chan et al., 2022; Bock et al., 2022). Techniques such as CRISPR-Cas9 (Jinek et al., 2012; Jiang & Doudna, 2017) knockout screens enable systematic perturbation of individual genes, but they are often resource-intensive and time-consuming. Given the vast number of protein-coding genes in the human genome (approximately 20,000), exhaustively testing all possible perturbations is infeasible (Abascal et al., 2018). Consequently, strategies that efficiently select the most informative experiments are essential to accelerate drug discovery while minimizing experimental costs.

Bayesian experimental design provides a principled framework for this challenge. In particular, Bayesian optimization (BO) offers a sample-efficient approach to identify genes whose perturbation maximizes desired cellular phenotypes. BO relies on a probabilistic surrogate model, such as a Gaussian process (Williams & Rasmussen, 2006) or a Bayesian neural network (Springenberg et al., 2016), to model the response surface, and an acquisition function to balance exploration of uncertain regions with exploitation of promising candidates (Frazier, 2018). While recent works have applied BO to gene perturbation design (Mehrjou et al., 2021; Lyle et al., 2023), they typically use generic, uni-modal gene representations (or embeddings) and do not fully leverage rich biological knowledge, limiting their performance. Integrating multimodal gene representations, which capture sequence, functional, and network-based information, can provide more informative representations and improve the efficiency of experimental selection.

Beyond richer gene representations, explicit biological priors can further guide experimental design. For example, gene set enrichment analysis (EA) identifies pathways that are statistically over-

---

*These authors contributed equally as first authors. Correspondence to: `tcui8@its.jnj.com`
†These authors contributed equally as senior (last) authors.

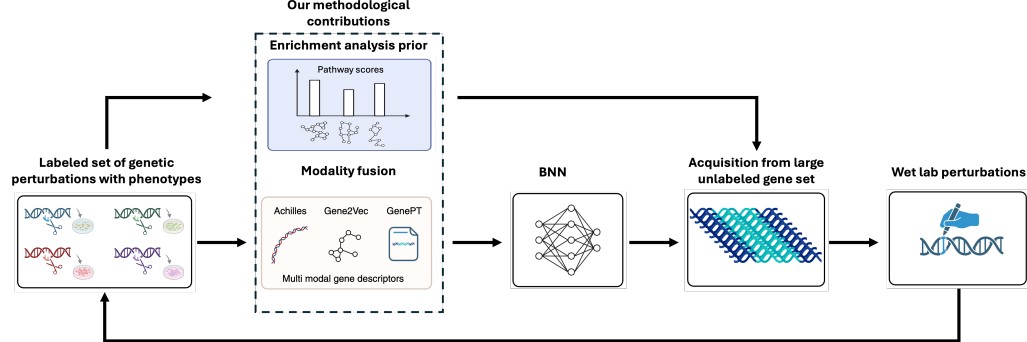

Figure 1: **BioBO pipeline for perturbation design**. We make two methodological innovations: (i). Fusion of gene modalities to improve surrogate modeling; (ii). Enrichment analysis on top of surrogate model predictions to strengthen gene acquisition via incorporating biological information.

represented among the top-performing genes, providing information on molecular mechanisms and potential high-value targets (Subramanian et al., 2005). However, conventional EA has two key limitations: (i) it lacks granularity, treating all genes within a pathway as equally promising, and (ii) it is purely exploitative, potentially biasing experiments toward well-characterized pathways while neglecting unexplored regions of the genome.

To address these limitations, we propose *Biology-Informed Bayesian Optimization* (BioBO), a framework that integrates multimodal gene representations and biological priors, such as enrichment analysis (Figure 1), into BO. BioBO helps balance exploration and exploitation, efficiently guiding experiments toward both well-characterized and underexplored genes. Together, these advances make BioBO a framework for efficient, interpretable, and effective experimental design, accelerating targeted discovery in genomic perturbation studies. Our key contributions are as follows.

1. We introduce multimodal gene embeddings, integrating multiple sources of biological information in the surrogate modeling to improve the designs of BO.

2. We demonstrate that the improvement of BO from multimodal embeddings is mainly from the improvement of surrogate model on regimes close to the optimum rather than on the entire data distribution.

3. We augment the acquisition function in BO using enrichment analysis within the theoretically principled $\pi$-BO (Hvarfner et al., 2022) framework. This approach incorporates prior biological knowledge while maintaining principled exploration–exploitation trade-off and provides interpretable insights into experimental design.

4. We empirically validate BioBO on established public benchmarks, showing that it outperforms conventional BO improves labeling efficiency by 25–40%, and identifies biologically coherent pathways with markedly stronger enrichment signals.

## 2 BACKGROUND AND NOTATION

### 2.1 NOTATION AND PROBLEM SETUP

We consider the task of optimizing a black-box function $f : \mathbb{G} \rightarrow \mathbb{R}$, which maps each gene $g \in \mathbb{G}$ represented by the set of integers or one-hot embeddings to a value $f(g) \in \mathbb{R}$ denoting the change of cell phenotype under the gene knockout, across the entire finite gene space $\mathbb{G}$ with $|\mathbb{G}| \approx 20,000$ (i.e., the number of protein-coding genes in human). Similar to (Lyle et al., 2023), we use biologically informed $d$-dimensional embeddings of genes, $\boldsymbol{X} : \mathbb{G} \rightarrow \mathbb{X}$, which maps each gene $g \in \mathbb{G}$ to a corresponding $d$-dimensional vector $\boldsymbol{X}(g) = \boldsymbol{x} \in \mathbb{X} \subseteq \mathbb{R}^d$ capturing the biological relationships with other genes. Moreover, the gene embeddings $\boldsymbol{X}$ construct a one-to-one mapping from $\mathbb{G}$ and contain the same number of distinct $d$-dimensional vectors as $\mathbb{G}$, i.e., $|\mathbb{X}| = |\mathbb{G}|$, so we use $f(\boldsymbol{x})$ and $f(g)$ interchangeably where $\boldsymbol{x}$ is the embedding of the gene $g$. Therefore, we define the optimization problem as follows

$$\boldsymbol{x}^* \in \arg\max_{\boldsymbol{x} \in \mathbb{X}} f(\boldsymbol{x}). \tag{1}$$

In practice, $f(\boldsymbol{x})$ is expensive to evaluate because it requires a CRISPR-Cas9 knockout experiment in the lab, and we would like to maximize $f(\boldsymbol{x})$ in an efficient manner by only evaluating a small number of points from $\mathbb{X}$. For this work, we do not perform wet-lab experiments ourselves; instead, we simulate the online BO loop by querying from a pool of genes with pre-measured phenotypes, as is standard practice in BO and Active Learning (AL) studies (Filstroff et al., 2021; Gupta et al., 2021; Li et al., 2024). While in practice, BO would operate on truly unlabeled genes, retrospective evaluation on fully labeled datasets is necessary to quantify and showcase the benefits of any BO or AL method.

## 2.2 BAYESIAN OPTIMIZATION

Bayesian optimization (BO) (Mockus, 1998; Frazier, 2018) is a model-based black-box function optimizer that employs a probabilistic model, e.g., Gaussian process (GP) (Williams & Rasmussen, 2006) or Bayesian neural network (BNN) (Springenberg et al., 2016), as a surrogate model. Specifically, BO optimizes $f$ from an initial experimental design $\mathcal{D}_1 = \{(\boldsymbol{x}_{1,i}, y_{1,i})\}_{i=1}^{M}$ and sequentially deciding on one or a batch (with size $B$) of new designs to label and form the data $\mathcal{D}_{n+1} = \mathcal{D}_n \cup \mathcal{B}_n$ with new labeled dataset $\mathcal{B}_n = \{(\boldsymbol{x}_{n,b}, y_{n,b})\}_{b=1}^{B}$ for the $n$-th iteration with $n \in \{1, \ldots, N\}$. At each iteration $n$, BO learns a probabilistic surrogate model $f_n \sim p(f_n|\mathcal{D}_n)$ to approximate the true function $f$, where $p(f_n|\mathcal{D}_n)$ is the posterior distribution of a GP or BNN given the labeled data. Using the predictive uncertainty from $p(f_n|\mathcal{D}_n)$, BO selects next designs by optimizing an acquisition function (AF), $\alpha_{p(f_n|\mathcal{D}_n)}(\boldsymbol{x})$, across the set of unlabeled data points.

Acquisition functions encapsulate the underlying utilities; therefore, they correspond to the trade-off between exploitation (using the current optimum from the surrogate model) and exploration (considering the uncertainty of the surrogate model). Popular choices of AF include Expected Improvement (EI) (Jones et al., 1998) and Upper Confidence Bound (UCB) (Srinivas et al., 2010). For instance, EI selects the next point $\boldsymbol{x}$ that maximizes the expected improvement:

$$\alpha_{p(f_n|\mathcal{D}_n)}^{\text{EI}}(\boldsymbol{x}) = \mathbb{E}[|f_n(\boldsymbol{x}) - y_n^*|^+] = Z\sigma_n(\boldsymbol{x})\Phi(Z) + \sigma_n(\boldsymbol{x})\phi(Z), \tag{2}$$

where $y_n^*$ is the best outcome observed so far, $Z = \frac{f_n(\boldsymbol{x}) - \mu_n(\boldsymbol{x})}{\sigma_n(\boldsymbol{x})}$ with $\mu_n(\boldsymbol{x})$ and $\sigma_n(\boldsymbol{x})$ representing the mean and variance of the posterior $p(f_n|\mathcal{D}_n)$ respectively, and $\phi(\cdot)$ and $\Phi(\cdot)$ are the PDF and CDF of standard Gaussian distribution. UCB is defined as:

$$\alpha_{p(f_n|\mathcal{D}_n)}^{\text{UCB}}(\boldsymbol{x}) = \mu_n(\boldsymbol{x}) + \kappa_n\sigma_n(\boldsymbol{x}), \tag{3}$$

where $\kappa_n$ is the user-specified parameter controlling the exploration–exploitation trade-off. Both EI and UCB provide a myopic strategy for determining informative designs with theoretical guarantees (Bull, 2011; Srinivas et al., 2010). Other popular myopic acquisition functions include Probability of Improvement (PI) (Jones, 2001), Thompson Sampling (TS) (Thompson, 1933), and DiscoBAX (Lyle et al., 2023). In this work, we mainly focus on using BNNs as surrogate models and UCB, EI, TS, and DiscoBAX as acquisition functions, similar to existing works on perturbation design (Mehrjou et al., 2021; Lyle et al., 2023); however, our work applies to other probabilistic models and myopic acquisition functions as well.

## 2.3 ENRICHMENT ANALYSIS

Enrichment analysis (EA) or over-representation analysis is a computational approach used to determine whether a set of genes associated with a specific biological process or pathway appears more often than expected by chance (Boyle et al., 2004; Khatri et al., 2012; Huang et al., 2009). Specifically, given a background gene set, e.g., all protein-coding human genes $\mathbb{G}$, and a subset $\mathbb{S} \subset \mathbb{G}$ of genes of interest, EA tests whether a pathway $i$, i.e., a predefined gene set $\mathbb{P}_i \subset \mathbb{G}$, with known biological function provided in pathway databases, such as Hallmark (Liberzon et al., 2015), is represented in $\mathbb{S}$ *statistically more frequently* than expected by chance.

EA has been widely used to design experiments in applications such as target prioritization and biomarker expansion (Katz et al., 2021; Zhao et al., 2022; Dai et al., 2022; Ramos et al., 2023; Ordóñez et al., 2024). Intuitively, if several desirable genes have been identified, EA can be applied

to discover the pathways enriched by those desirable genes. Therefore, other untested genes in those significantly enriched pathways would construct a good candidate set for the next round of experiments. The significantly enriched pathways serve as a biologically informed prioritization framework for designing experiments, allowing us to target molecular processes where the desirable genes are most likely to be. This approach ensures that experimental interventions are focused on high-value genes within the biological network, thereby increasing the likelihood of eliciting interpretable system-level responses while reducing experimental redundancy.

Although EA serves as a well-established, biologically informed experimental design framework, it contains two major shortcomings:

1. Lack of granularity: EA can prioritize pathways; however, all untested genes in the same pathway are equally likely. This can still construct a huge pool if the significantly enriched pathway is large.

2. Lack of exploration: EA-based experimental design is a pure exploitation process and has potential bias toward known biology. The significantly enriched pathway would be more exploited by selecting more genes from it, and non-significant pathways will never be explored.

In this work, we propose a principled approach to combine the BO-based and EA-based experimental design framework to equip BO with extensive domain information in biology from EA and equip EA with granularity and exploration from BO.

## 3 METHOD: BIOLOGY-INFORMED BAYESIAN OPTIMIZATION

### 3.1 SURROGATE MODELLING WITH MULTIMODAL GENE REPRESENTATIONS

We first improve BO by improving the surrogate modeling. Specifically, we propose to use multi-modal gene embeddings rather than the uni-modal embeddings used in the existing gene perturbation design literature (Mehrjou et al., 2021; Lyle et al., 2023). We consider the following two extra gene embeddings that are effective in many gene-level tasks (Yang et al., 2022; Chen & Zou, 2025):

1. Gene2Vec (Du et al., 2019), $x^{g2v}$: gene embeddings encode gene-gene relations defined in gene ontology (Ashburner et al., 2000) learned with self-supervised learning;

2. GenePT (Chen & Zou, 2025), $x^{GenePT}$: ChatGPT embeddings of genes based on the literature.

We use Bayesian Neural Networks (BNNs) as surrogate models similar to previous works (Mehrjou et al., 2021; Lyle et al., 2023), and we concatenate the original gene embedding $x$ with the gene embeddings from the above-mentioned modalities as the input of a BNN, i.e., $f([x, x^{g2v}, x^{GenePT}])$. We also explore a latent-space fusion strategy, which learns a joint representation integrating the heterogeneous biological modalities in latent space either via concatenation or using cross-attention.

In Section 4.3, we design a comprehensive analysis to study relations between the performance of BO and surrogate models to reveal reasons behind the benefits of multimodal fusion in BO settings.

### 3.2 AUGMENTED ACQUISITION FUNCTION WITH ENRICHMENT ANALYSIS

Vanilla BO ignores prior beliefs about the optimum's location, overlooking valuable knowledge that could enhance the search. We mainly focus on $\pi$BO (Hvarfner et al., 2022), a principled generalization of the acquisition function to incorporate prior beliefs about the location of the optimum in the form of probability distributions $\pi(x)$. Specifically, for acquisition function $\alpha_{p(f_n|\mathcal{D}_n)}(x)$, the corresponding augmented acquisition function is:

$$\pi\alpha_{p(f_n|\mathcal{D}_n)}(x) = \alpha_{p(f_n|\mathcal{D}_n)}(x)\pi_n(x)^{\frac{\beta}{L_n}}, \tag{4}$$

where $\beta$ is a hyperparameter set by the user (see a sensitivity analysis of $\beta$ in Appendix C), reflecting their confidence in $\pi_n(x)$, and $L_n$ is the number of labeled data so far. This reflects the intuition that, as the optimization progresses, we should increasingly trust the surrogate model over the prior, as BO will likely have enough data to reach the optimum confidently. This also comes with theoretical properties described in the next section.

In this work, we propose to augment the acquisition function with the prioritization results from enrichment analysis as a prior within the $\pi$BO framework. Enrichment analysis comes with statis-

tical hypothesis tests: under the null hypothesis $\mathcal{H}_0$, that genes in $\mathbb{S}$ are sampled uniformly from $\mathbb{G}$, the probability of observing at least $|\mathbb{S} \cap \mathbb{P}_i|$ overlaps follows the upper tail of the hypergeometric distribution; therefore, we can compute the p-value with

$$p(\mathbb{P}_i) = \sum_{i=|\mathbb{S} \cap \mathbb{P}_i|}^{\min(|\mathbb{P}_i|,|\mathbb{S}|)} \binom{|\mathbb{P}_i|}{i} \binom{|\mathbb{G}| - |\mathbb{P}_i|}{|\mathbb{S}| - i} \Big/ \binom{|\mathbb{G}|}{|\mathbb{S}|} , \qquad (5)$$

and multiple hypothesis testing across all pathways is controlled via Bonferroni correction (Haynes, 2013) to derive the adjusted p-value, $p^{\mathrm{adj}}(\mathbb{P}_i)$. One can also compute the odds ratio, $o(\mathbb{P}_i)$, from the EA results by constructing the contingency table, and a high $o(\mathbb{P}_i)$ (e.g., $o(\mathbb{P}_i) > 1$) indicates that $\mathbb{P}_i$ is over-represented in $\mathbb{S}$ compared to random. Chen et al. (2013) propose to combine the p-value and odds ratio to evaluate the overall representativeness with $c(\mathbb{P}_i) = -o(\mathbb{P}_i) \log p(\mathbb{P}_i)$, which will be used to design the biologically informed prior $\pi_n(\boldsymbol{x})$ at each iteration.

At each iteration $n$, we rank labeled genes according to their labels (i.e., change of phenotype under the gene knockout). We consider the top-k (we use top-10% in this paper and report the results with di) genes as the genes of interest, i.e., $\mathbb{S}_n$, and use enrichment analysis (Chen et al., 2013) to find top enriched pathways, ranked by the combined score $c(\mathbb{P}_i)$. We additionally provide sensitivity analysis of BioBO to this choice of $k$ in Appendix H. If one unlabeled gene is within the top pathway, we increase the probability of selecting the gene in the acquisition function. Specifically, we define the probability of selecting an unlabeled gene $\boldsymbol{x}$ as follows:

$$s_n(\boldsymbol{x}) = \mathrm{logit}(\frac{1}{U_n}) + \frac{1}{t} \mathbf{agg}_{\{\mathbb{P}_i | \boldsymbol{x} \in \mathbb{P}_i, p_n^{\mathrm{adj}}(\mathbb{P}_i) < 0.05\}} [c_n(\mathbb{P}_i)], \quad \pi_n(\boldsymbol{x}) = \frac{e^{s_n(\boldsymbol{x})}}{\sum_{\boldsymbol{x}} e^{s_n(\boldsymbol{x})}}, \qquad (6)$$

where $U_n$ is the number of unlabeled genes at iteration $n$ and $\mathbf{agg}[\cdot]$ is a set aggregation operation that summarizes the combined score $c_n(\cdot)$ at iteration $n$ across all significant pathways (with adjusted p-value $p_n^{\mathrm{adj}}(\mathbb{P}_i) < 0.05$) that contains the unlabeled gene $\boldsymbol{x}$. We use **mean** operation in the paper to measure the averaged representativeness in all significant pathways. We also explore the **max** operation in Appendix C, which shows benefits as well. The hyperparameter temperature $t$ controls the level of information that we keep from the enrichment analysis. When $t = \infty$, $\pi(\boldsymbol{x})$ reduces to a uniform distribution, and EA will be ignored. We use $t = 0.1$ in all experiments.

### 3.2.1 THEORETICAL PROPERTIES

BioBO comes with the same *no-harm guarantee* as the original $\pi$BO (Hvarfner et al., 2022), because of the decaying effect of the prior in Eq.4 when employed with myopic AFs (all AFs used in this paper). For instance, when paired with the EI, we can prove that the regret, $\mathcal{L}_n(\mathrm{BioEI}_n)$, to the optimum at iteration $n$ of the BioEI strategy, i.e., using EI in Eq.4, can be bounded by the regret of the corresponding EI strategy, $\mathcal{L}_n(\mathrm{EI}_n)$, using the Theorem 1 of Hvarfner et al. (2022) as following:

$$\mathcal{L}_n(\mathrm{BioEI}_n) \leq C_{\pi,n} \mathcal{L}_n(\mathrm{EI}_n), \quad C_{\pi,n} = \left( \frac{\max_{\boldsymbol{x}} \pi_n(\boldsymbol{x})}{\min_{\boldsymbol{x}} \pi_n(\boldsymbol{x})} \right)^{\frac{\beta}{L_n}} . \qquad (7)$$

For detailed conditions and proofs of the above Theorem, please refer to the original $\pi$BO paper (Hvarfner et al., 2022). Therefore, we have the *no-harm guarantee* that the regret of the BioEI strategy is asymptotically equal to the regret of the EI strategy:

$$\mathcal{L}_n(\mathrm{BioEI}_n) \sim \mathcal{L}_n(\mathrm{EI}_n), \qquad (8)$$

which indicates that BioEI is robust against errors and biases from the enrichment analysis.

## 4 EXPERIMENTS

### 4.1 GENEDISCO DATASETS

**Datasets** We use five genome-wide CRISPR assays from the GeneDisco dataset (Mehrjou et al., 2021) and present the analysis for the two most widely-used datasets from literature (IFN-$\gamma$ and IL-2) in the main text, while the same analysis for others is shown in Appendix. We use the Achilles gene descriptor, i.e., gene embedding $\boldsymbol{X}$, from GeneDisco. Although GeneDisco includes other two

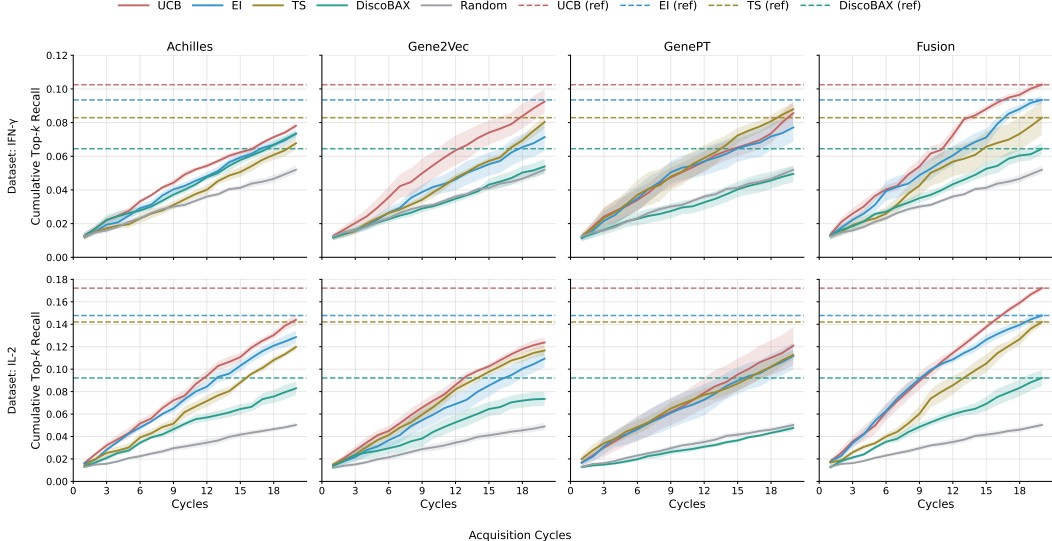

Figure 2: **Performance across single modalities (Achilles, Gene2Vec, GenePT) and their Fusion on IFN-$\gamma$ (top) and IL-2 (bottom)**. Row-wise dashed lines indicate the Fusion value at the final cycle (20) for UCB, EI, TS, and DiscoBAX to aid comparison. We observe that BO with Fusion is better than BO with any single modality.

gene descriptors, CCLE and STRING, only Achilles is informative to predict the cell phenotypes, as shown in (Mehrjou et al., 2021; Lyle et al., 2023) and Appendix Section E.1; therefore, we focus on Achilles from GeneDisco. For richer gene representations, we go beyond unimodal Achilles and include two additional embeddings: Gene2Vec and GenePT (in Section 3.1) to leverage multimodal genetic descriptors. For additional details on datasets and descriptors, see Appendix Section A.

**Measure the performance of BO**  We use Cumulative Top-k Recall to measure the ability of a method to identify the top gene perturbations as those in the top percentile of the experimentally measured phenotypes following Lyle et al. (2023).

**Measure the performance of surrogate models**  We evaluate surrogate models on a separate test set using LL (log-likelihood) for the quality of predictive distribution and RMSE (Root Mean Squared Error) for the prediction accuracy. Moreover, we calculate LL and RMSE on subsets of the test data that are close to the optimum, e.g., LL@top-10% represents the LL on the test data points whose labels are within the top 10%, to evaluate the model performance near the maximum.

**Baselines**  For surrogate models, we use a BNN in (Lyle et al., 2023), using Achilles, Gene2Vec, GenePT, and Fusion (i.e., the fusion of three modalities). We use UCB, EI, TS, DiscoBAX as acquisition functions, as well as augmented acquisition functions, BioUCB, BioEI, and BioTS, with biological priors from enrichment analysis using Gene Ontology (GO) (Ashburner et al., 2000) and Hallmark (HM) (Liberzon et al., 2015) databases. We run each experiment with 7 different seeds.

### 4.2 EXPLORING EFFECTS OF USING MULTIMODAL GENE REPRESENTATIONS IN BO

First, we study the effects of using multimodal gene representations, i.e., the Fusion, in surrogate models. Figure 2 shows the cumulative top-k recall of different acquisition functions at each cycle of the experimental design. We observe that all BO acquisition functions are better than random, especially UCB, and BO saves the labeling efforts 25%-75% compared with random, which indicates the benefits of BO in experimental design. Moreover, we observe that using surrogate models with the Fusion is always better than using single-modal surrogate models, with labeling effort saving ranging from 4% to 40%. The best-performing model is using the Fusion with UCB. We also observe that DiscoBAX (Lyle et al., 2023) is worse than existing standard acquisition functions[1], and hence we remove DiscoBAX in the subsequent experiments. In addition, as detailed in Appendix F,

---

[1]This observation is consistent with an issue reported by the DiscoBAX authors in their official GitHub repository (Issue #3), noting that the originally reported performance was affected by an implementation bug.

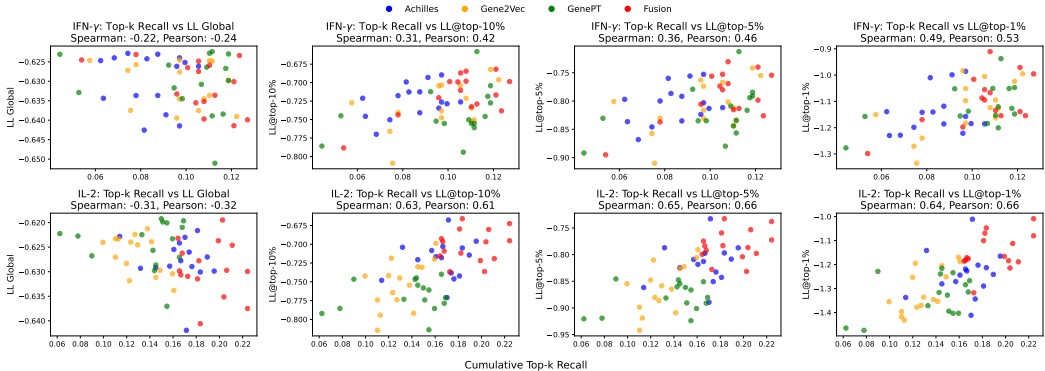

Figure 3: **Relations between performance of BO and the surrogate model**. We observe that Fusion (red) does not improve the surrogate model **globally** (LL global, first column). However, it improves on data points that are **near optimum** (LL@top-1% to LL@top-10%), which explains the improvement on BO results (top-k recall). Specifically, the top-k recall of BO is more correlated with **local LL** than global LL, measured by both Spearman and Pearson correlation.

The latent-space fusion strategies further improve BO performance over simple concatenation-based fusion, highlighting the advantage of integrating heterogeneous modalities more effectively.

### 4.3 ANALYZING RELATIONS BETWEEN PERFORMANCE OF BO AND SURROGATE MODELS

Observing the benefits of Fusion in BO from Figure 2, we further analyze why using Fusion in the surrogate model improves BO. One intuitive hypothesis is that: *a more expressive multimodal gene representation improves the predictive distribution of the surrogate model, which leads to better Bayesian Optimization*. We test this hypothesis by estimating the correlation between the performance of BO and surrogate model. Specifically, we divide the dataset into training and testing: we run BO loops on the training set and measure the performance of BO (cumulative top-k recall), and we measure the performance of the surrogate model on the test set. We plot the performance of BO (cumulative top-k recall) and the surrogate model (test LL) in Figure 3.

We find that the correlation between cumulative top-k recall and LL is negative (first column in Figure 3), meaning a higher LL does not lead to a better BO. Although counterintuitive, it is consistent with the conclusions from Foldager et al. (2023). In BO, however, the surrogate is primarily used to estimate the relative ordering of high-value candidates and to locate the local optimum near top-performing genes, rather than to achieve high global predictive accuracy. As also noted in Foldager et al. (2023), global likelihood therefore has limited influence on the acquisition function. Thus, even if fusion does not improve global likelihood, it can still enhance BO performance when it sharpens the surrogate locally. We observe precisely this effect: the predictive distribution of the surrogate model improves **near optimum** (red dots are higher than others on average: second, third, and fourth columns in Figure 3), which is positively correlated with the BO performance significantly, with Spearman correlation ranging from $0.31$ to $0.49$ for IFN-$\gamma$ and being around $0.64$ for IL-2. Moreover, we observe the highest Spearman correlation of cumulative top-k recall is with LL@top-1% on 4/5 datasets (see the results for other three datasets in Appendix Section E.3). Therefore, we conclude that: *multimodal gene embedding improves the predictive distribution of the surrogate model **near optimum**, which leads to a better Bayesian optimization.*

### 4.4 EXPLORING EFFECTS OF COMBINING ENRICHMENT ANALYSIS IN BO

Here, we study the benefits of combining enrichment analysis with BO using the proposed BioBO framework in design experiments. First, we analyze if the prior distribution in Eq.6, constructed from results of enrichment analysis, is beneficial in experimental design, i.e., using a model-free approach. We select genes with the highest prior probabilities (greedy selection) in Eq.6. Figure 4(a) shows that using both Gene Ontology and Hallmark as the pathway database for enrichment can improve the design compared with random selection of genes, demonstrating the potential of enrichment analysis to inform experimental design. However, this approach is purely exploitative.

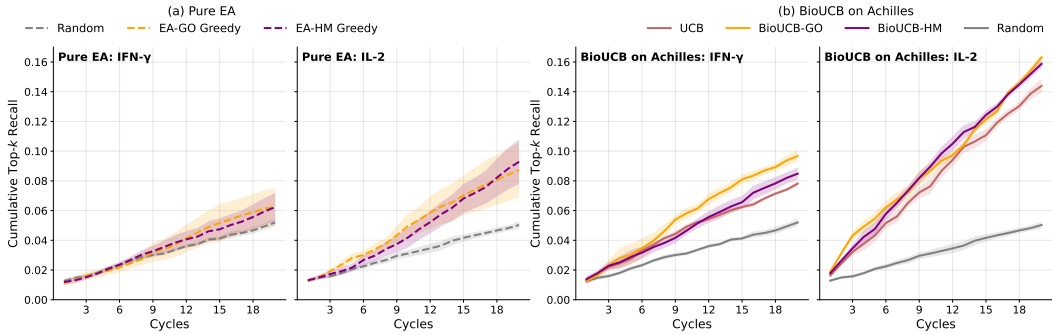

Figure 4: **Performance of pure EA and BioUCB on Achilles.** (a): Pure EA on IFN-$\gamma$ and IL-2. We observe that pure EA provides better designs than random. (b): BioUCB on Achilles for IFN-$\gamma$ and IL-2. We observe that BioUCB provides better designs than UCB and pure EA.

Table 1: **Cumulative top-k recall with standard error of each acquisition function on different datasets**. We observe that BioBO achieves the best performance on 23/24 different settings, and BioUCB-HM with surrogate function using fused features achieves the best performance for both IFN-$\gamma$ and IL-2. The best performance (with the smallest standard error) is bold.

| Phenotype: IFN-$\gamma$ | Fusion | Achilles | GenePT | Gene2Vec |
|---|---|---|---|---|
| EI | 0.093 (0.001) | 0.072 (0.001) | 0.077 (0.004) | 0.071 (0.006) |
| BioEI-GO (ours) | **0.098 (0.000)** | **0.085 (0.000)** | 0.095 (0.005) | 0.079 (0.004) |
| BioEI-HM (ours) | 0.096 (0.001) | 0.076 (0.001) | **0.096 (0.007)** | **0.079 (0.002)** |
| TS | 0.083 (0.001) | 0.068 (0.001) | 0.088 (0.002) | 0.073 (0.002) |
| BioTS-GO (ours) | 0.095 (0.001) | 0.073 (0.000) | **0.097 (0.004)** | **0.095 (0.005)** |
| BioTS-HM (ours) | **0.097 (0.001)** | **0.097 (0.005)** | 0.093 (0.005) | 0.081 (0.004) |
| UCB | 0.100 (0.001) | 0.077 (0.001) | 0.086 (0.004) | 0.093 (0.005) |
| BioUCB-GO (ours) | 0.102 (0.001) | **0.098 (0.002)** | 0.092 (0.005) | 0.098 (0.002) |
| BioUCB-HM (ours) | **0.109 (0.001)** | 0.085 (0.003) | **0.101 (0.001)** | **0.103 (0.004)** |
| Random | 0.050 (0.001) | 0.050 (0.001) | 0.050 (0.001) | 0.050 (0.001) |
| Phenotype: IL-2 | Fusion | Achilles | GenePT | Gene2Vec |
| EI | 0.148 (0.002) | 0.130 (0.003) | 0.107 (0.005) | 0.109 (0.002) |
| BioEI-GO (ours) | 0.147 (0.003) | **0.138 (0.003)** | 0.107 (0.005) | **0.115 (0.002)** |
| BioEI-HM (ours) | **0.153 (0.002)** | 0.130 (0.003) | 0.107 (0.005) | 0.109 (0.002) |
| TS | 0.142 (0.001) | 0.119 (0.001) | 0.113 (0.014) | 0.113 (0.002) |
| BioTS-GO (ours) | 0.147 (0.003) | **0.142 (0.002)** | 0.119 (0.011) | 0.119 (0.001) |
| BioTS-HM (ours) | **0.153 (0.002)** | 0.123 (0.004) | **0.139 (0.013)** | **0.124 (0.002)** |
| UCB | 0.174 (0.001) | 0.143 (0.003) | 0.118 (0.011) | 0.123 (0.000) |
| BioUCB-GO (ours) | 0.169 (0.001) | 0.158 (0.001) | 0.131 (0.008) | **0.133 (0.002)** |
| BioUCB-HM (ours) | **0.178 (0.001)** | **0.163 (0.001)** | **0.138 (0.012)** | 0.127 (0.000) |
| Random | 0.049 (0.001) | 0.048 (0.001) | 0.049 (0.001) | 0.046 (0.002) |

Next, we combine the enrichment analysis prior with the acquisition function in BO, i.e., the model-based BioBO approach, thus balancing eploitation-exploration trade-off explained in Section 3.2. We observe that adding the enrichment analysis prior can improve the labeling efficiency over BO with the corresponding acquisition function without the prior. Specifically, the enrichment analysis prior improves the labeling efficiency of UCB by 20% with Achilles gene embedding on optimizing IFN-$\gamma$. We show the cumulative top-k recall of all experiments in Table 12, where we observe that the prior from enrichment analysis can improve the original acquisition function most of the time (23/24 cases). The best performance is achieved by BioUCB using the Hallmark database for building the enrichment prior with fused gene embeddings in both IFN-$\gamma$ and IL-2.

## 4.5 INTERPRETABILITY OF DESIGNS

In this section, we conduct enrichment analysis using the Hallmark dataset to provide biological interpretations of selected genes by BO. We compare two models on the IFN-$\gamma$ dataset: the baseline

Table 2: **Enrichment analysis results of designs from existing method and BioBO**. We observed that our BioUCB-HM with multimodal gene embedding shows significantly stronger enrichment signals compared to existing approach (BO with UCB).

| Phenotype: IFN-$\gamma$; Feature: Achilles; Acquisition: UCB | | | | |
|---|---|---|---|---|
| Pathway | Overlap | Adjusted p-value | Odds Ratio | Combined Score |
| MYC_TARGETS_V1 | 32/200 | $2.71 \times 10^{-13}$ | 7.25 | 237.22 |
| E2F_TARGETS | 22/200 | $5.10 \times 10^{-6}$ | 4.32 | 66.18 |
| DNA_REPAIR | 14/150 | $3.87 \times 10^{-3}$ | 3.45 | 28.51 |
| G2M_CHECKPOINT | 15/200 | $1.79 \times 10^{-2}$ | 2.7 | 17.42 |
| Phenotype: IFN-$\gamma$; Feature: Fusion; Acquisition: BioUCB-HM | | | | |
| Pathway | Overlap | Adjusted p-value | Odds Ratio | Combined Score |
| MYC_TARGETS_V1 | 187/200 | $4.98 \times 10^{-247}$ | 766.31 | $4.37 \times 10^5$ |
| E2F_TARGETS | 48/200 | $1.07 \times 10^{-16}$ | 5.92 | 235.98 |
| G2M_CHECKPOINT | 40/200 | $3.93 \times 10^{-11}$ | 4.52 | 120.41 |
| MYC_TARGETS_V2 | 18/58 | $2.90 \times 10^{-8}$ | 7.66 | 151.48 |

UCB + Achilles and our method BioUCB-HM + Fusion. Table 2 shows that BioUCB-HM produces markedly stronger enrichment signals in pathways closely tied to IFN-$\gamma$ regulation in T cells. While UCB identifies relevant pathways such as MYC_TARGETS_V1 and E2F_TARGETS with modest overlaps (32/200 and 22/200) and adjusted p-values in the range of $10^{-13}$ to $10^{-2}$, BioUCB-HM shows stronger enrichment signals compared to UCB. For example, MYC_TARGETS_V1 reaches an extraordinary overlap of 187/200 genes with an adjusted p-value of $4.98 \times 10^{-247}$, yielding a combined score over 1,000-fold higher than UCB. Similarly, other critical pathways such as E2F_TARGETS and G2M_CHECKPOINT not only remain significant but also demonstrate substantially higher overlaps and more robust statistics under BioUCB-HM, while BioUCB-HM further uncovers MYC_TARGETS_V2, missed entirely by UCB. From a biological perspective, these pathways are central regulators of cell growth, proliferation, and metabolism. MYC drives effector T cell proliferation but can restrain differentiation, so its inhibition is consistent with enhanced IFN-$\gamma$ production (Melnik et al., 2019). Likewise, targeting E2F and G2M checkpoint regulators reduces proliferation pressure and shifts T cell programming toward cytokine output, while DNA repair mechanisms also intersect with stress responses in activated T cells (Ren et al., 2002). The observation that knockout of genes in these pathways increases IFN-$\gamma$ log fold change supports the idea that restraining proliferative and metabolic circuits frees T cells to mount stronger effector responses. Thus, BioUCB-HM not only outperforms UCB quantitatively but also pinpoints biologically meaningful regulatory axes—MYC, E2F, and G2M—that provide a mechanistic rationale for boosting IFN-$\gamma$ production in T cells. Further analysis of underexplored biologically novel genes prioritized by BioBO is detailed in Appendix N alongside the biological mechanistic interpretability of these genes.

Beyond IFN-$\gamma$, we additionally evaluate BioBO on a second immune-cell perturbation IL-2 dataset and observe qualitatively similar interpretability gains: BioUCB-HM consistently produces markedly stronger and more biologically coherent enrichment signals than baseline UCB. Full results and pathway-level statistics are provided in Appendix I .

## 4.6 COMPUTATIONAL EFFICIENCY OF BIOBO

Runtime per iteration of BioBO is comparable to existing BO methods. We report detailed runtimes in Appendix G. The choice of 20 acquisition cycles (selecting 400 genes with 20 genes per cycle) follows exactly the experimental protocol established in (Mehrjou et al., 2021; Lyle et al., 2023), ensuring comparability. The total of 400 perturbations selected by 20 iterations corresponds to less than 5% of the typical gene pool, aligning with realistic experimental budgets in high-throughput CRISPR screens (Mehrjou et al., 2021; Lyle et al., 2023). Thus, BioBO maintains its fastness from a practical standpoint and identifies high-value perturbations more efficiently compared to baseline methods.

## 5 OTHER RELATED WORKS

**Exploiting external knowledge for drug discovery**   Incorporating external knowledge has recently been studied extensively in drug discovery. Neural network architectures that have domain-specific inductive biases (Cui et al., 2022a; Yazdani-Jahromi et al., 2025; Moskalev et al., 2025) and explicit Bayesian priors constructed from existing knowledge (Cui et al., 2022b; Skok Gibbs et al., 2024; Cui et al., 2025; de la Fuente Cedeño et al., 2025) have been proposed for various purposes. In BO, external knowledge can be elicited from the feedback of human experts through preference learning and used in BO (Mikkola et al., 2020; Adachi et al., 2023) when the explicit knowledge is challenging to obtain. However, when the external knowledge on the input space over the potential candidates is ready, it can be either treated as a constraint (Hernández-Lobato et al., 2015; Adachi et al., 2022) or a prior belief (Souza et al., 2021; Hvarfner et al., 2022; Cissé et al., 2024), and our BioBO fits within this framework.

**Experimental design in drug discovery**   Many drug discovery and design applications use experimental design to speed up the process. Active learning, a framework that finds the most informative unlabeled datapoints to label for improving the model, has been applied to molecular property prediction (Neporozhnii et al., 2025; Masood et al., 2025), Perturb-seq experiments  (Zhang et al., 2023; Huang et al., 2024), and genomics CRISPR assays (Mehrjou et al., 2021).  Active learning uses the information gain of the probabilistic surrogate model to guide the selection, such as BALD (Houlsby et al., 2011) and EPIG (Smith et al., 2023); therefore, it is an exploration-only process. On the other hand, BO trades off between exploration and exploitation to query the most informative unlabeled datapoints to the optimum.  BO has been applied to bio-sequence optimization by combining with deep generative models, including small-molecular and protein sequences (Gómez-Bombarelli et al., 2018; Stanton et al., 2022; Gruver et al., 2023; Ramchandran et al., 2025), as well as on genomics CRISPR assays (Pacchiano et al., 2023; Lyle et al., 2023).  Recently, large language model (LLM) based agents have shown potential in experimental design by leveraging the rich background knowledge and reasoning capabilities (Lee et al., 2024; Roohani et al., 2025), and enrichment analysis has been shown to be an important tool in the multi-agent system (Hao et al., 2025).  Different from heuristic designs with LLM, we focus on a well-principled Bayesian experimental design framework.

## 6 CONCLUSION

We introduce BioBO, a biology-informed BO framework for perturbation design, combining standard BO with multimodal gene representations and enrichment analysis to guide experimental prioritization. Our theoretical analysis establishes a no-harm guarantee when integrating biological priors from enrichment analysis, ensuring robustness to noisy or biased pathway information. Empirical results on the GeneDisco datasets demonstrate substantial gains in sample efficiency, with BioBO outperforming traditional BO methods and enrichment-only strategies. By fusing principled optimization with domain-specific biological insights, BioBO enables more efficient discovery of high-value perturbations, reducing experimental costs. We also analyze failure cases, showing that when an incorrect or biologically mismatched pathway resource is used, the enrichment prior becomes uninformative and BioBO gracefully reduces to the underlying surrogate model (see Appendix J). Finally, we evaluate BioBO in realistic settings where some embedding modalities are unavailable, showing that simple KNN-imputation preserves strong performance and that multimodal fusion continues to outperform single-modality surrogates (Appendix K). BioBO can also integrate multiple enrichment sources simultaneously, and ensemble priors consistently match or outperform individual databases (Appendix M). Looking forward, this approach provides a foundation for integrating broader biological knowledge sources—such as single-cell profiles and literature-derived embeddings—into experimental design frameworks, paving the way for faster and more targeted advances in genomics and therapeutic discovery.

## REPRODUCIBILITY STATEMENT

To facilitate reproducibility, we provide data description in Section A and implementation details, including choice of computational platform and model hyperparameters, in Section B. Code will be released upon acceptance.

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

## A  Data Description

GeneDisco contains three different embeddings: Achilles (dependency score of genetic intervention across cancer cell lines) (Dempster et al., 2019), STRING (protein-protein interactions) (Szklarczyk et al., 2021), and CCLE (quantitative proteomics information from cancer cell lines) (Nusinow et al., 2020), which are available for 17,655, 17,972, and 11,943 genes. We also consider two gene embeddings: Gene2Vec and GenePT, which are available for 23,940 and 61,287 genes. In order to remove the effect of the different missingness levels of each gene embedding, we use the 10,556 genes that have all five embeddings.

GeneDisco also contains 5 datasets from genome-wide CRISPR assays: IFN-$\gamma$, IL-2 (the log fold change of Interferon-$\gamma$ and Interleukin-2 production in primary human T cells (Schmidt et al., 2021)), Tau (Tau protein assay (Sanchez et al., 2021)), NK (Leukemia assay with NK cells (Zhuang et al., 2019)), and Sars-Covid2 (SARS-CoV-2 assay from (Zhu et al., 2021)). we consider an intersection of genes with all modalities and each assay.

## B  Experimental Details

**Device details**   All experiments were run on Debian GNU/Linux 10 (buster) with Python 3.10.16, PyTorch 2.6.0, and CUDA 12.8. Training and inference used two NVIDIA L4 GPUs (each with 24 GB VRAM). The host machine had an AMD EPYC 7R13 processor with 192 hardware threads and 80 GB of system memory. Computations used 64-bit floating-point precision where required by the Bayesian layers.

**Hyperparameters**   Unless noted, the BNN surrogate is a Monte Carlo (MC) dropout neural network using a *2*-layer MLP having a hidden width 64 and ReLU activations with dropout rate 0.5. We optimize BNNs with Adam (learning rate $\eta = 0.001$, weight decay $\lambda = 0.0001$) for up to 200 epochs with early stopping (patience 30) on a 10% validation split; batch size was 256. The mean and variance of the posterior distribution used in acquisition functions are estimated from 100 samples collected by MC dropout during testing. For modality fusion, we concatenated L2-normalized embeddings (Achilles, Gene2Vec, GenePT; and where used, CCLE/STRING). Acquisition functions followed standard definitions for UCB (trade-off $\kappa_n = 1$), EI ($\xi = 0$), and TS; biology-informed variants added enrichment weights from GO or Hallmark with temperature coefficient $t = 0.1$ and $\beta = 1$ for IFN-$\gamma$ and $\beta = 0.1$ for IL-2.

**Reproducibility and error bars**   For every dataset–modality–acquisition setting we ran **seven** independent random seeds. Plotted curves report the mean across seeds; shaded bands show $\pm$ s.e.m. (standard error of the mean). Final-cycle bar plots likewise report mean $\pm$ s.e.m. Each BO iteration in our experiments acquires a batch of 20 genes ($B = 20$) rather than a single gene, reflecting a realistic experimental design.

## C  Sensitivity Analysis

We analyze the sensitivity of the BO results w.r.t. $\beta$ in Eq.4 for both **mean** and **max** aggregation operation on IFN-$\gamma$. We observe that both **mean** and **max** aggregation can bring the benefits of EA into BO. While performance varies across extreme $\beta$ values, we observe that $\beta$ in the range 1-5 generally yields the best performance across acquisition functions and datasets (Appendix C). This is also expected as $\beta$ controls the extent to which the enrichment prior influences the acquisition score. Small $\beta$ hence effectively removes the influence of biological structure, thus yielding poorer performance compared to using moderate $\beta$ in the range 1-5.

| | Phenotype: IFN-$\gamma$; Feature: Achilles | | | | | |
|---|---|---|---|---|---|---|
| Acquisition | $\beta = 0.01$ | $\beta = 0.05$ | $\beta = 0.1$ | $\beta = 0.5$ | $\beta = 1$ | $\beta = 5$ |
| BioEI-GO (mean) | 0.0756 | 0.0756 | 0.0756 | 0.0756 | **0.0852** | 0.0846 |
| BioEI-GO (max) | 0.0848 | 0.0770 | 0.0770 | 0.0856 | 0.0873 | **0.0969** |
| BioEI-HM (mean) | 0.0756 | 0.0756 | 0.0756 | 0.0756 | **0.0763** | 0.0710 |
| BioEI-HM (max) | 0.0756 | 0.0756 | 0.0756 | 0.0760 | **0.0764** | 0.0743 |
| BioUCB-GO (mean) | 0.0850 | 0.0891 | **0.0984** | 0.0978 | 0.0975 | 0.0944 |
| BioUCB-GO (max) | 0.0877 | 0.0919 | **0.0956** | 0.0919 | 0.0750 | 0.0731 |
| BioUCB-HM (mean) | 0.0726 | 0.0731 | 0.0754 | 0.0816 | **0.0848** | 0.0833 |
| BioUCB-HM (max) | 0.0752 | 0.0747 | 0.0754 | 0.0764 | **0.0850** | 0.0836 |

## D  LLMs USAGE

Large Language Models (LLMs) were used to assist word choice and improve grammar.

## E  SUPPLEMENTARY EXPERIMENTAL RESULTS

### E.1  CCLE AND STRING MODALITIES IN GENEDISCO

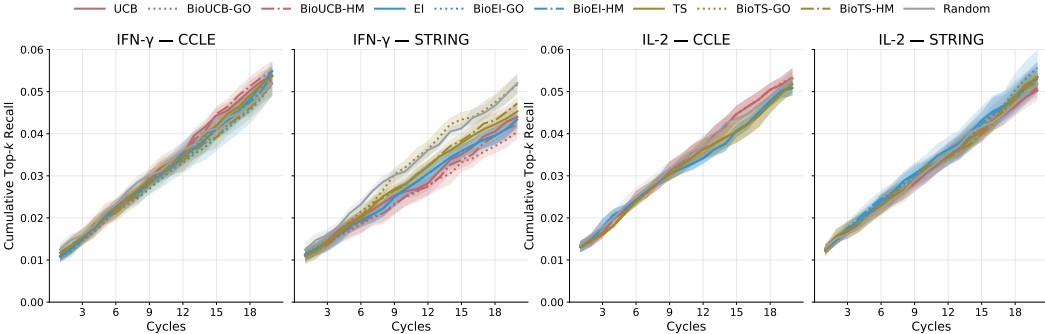

Figure 5: CCLE and STRING modalities across datasets. Panels (left→right): IFN-$\gamma$—CCLE, IFN-$\gamma$—STRING, IL-2—CCLE, IL-2—STRING. Curves show base acquisitions **UCB/EI/TS** (solid), biology-informed variants **BioUCB/BioEI/BioTS** with **GO** (dotted) and **HM** (dash–dot) in the same family color, plus **Random** (gray). Shaded ribbons denote mean $\pm$ s.e.m.

In this section, we studied two other modalities, CCLE and STRING from GeneDisco in Figure 5. We observe that both CCLE and STRING yield substantially lower absolute recall compared to the Achilles, Gene2Vec, and GenePT features. Moreover BO is similar to random acquisition using these two embeddings, which indicates that both CCLE and STRING are less informative to predict the selected phenotype. We exclude them from the main paper and report them here for completeness. Even so, biology-informed variants provide modest, consistent gains over their bases—particularly at smaller budgets.

## E.2 BO RESULTS FOR IFN-γ AND IL-2

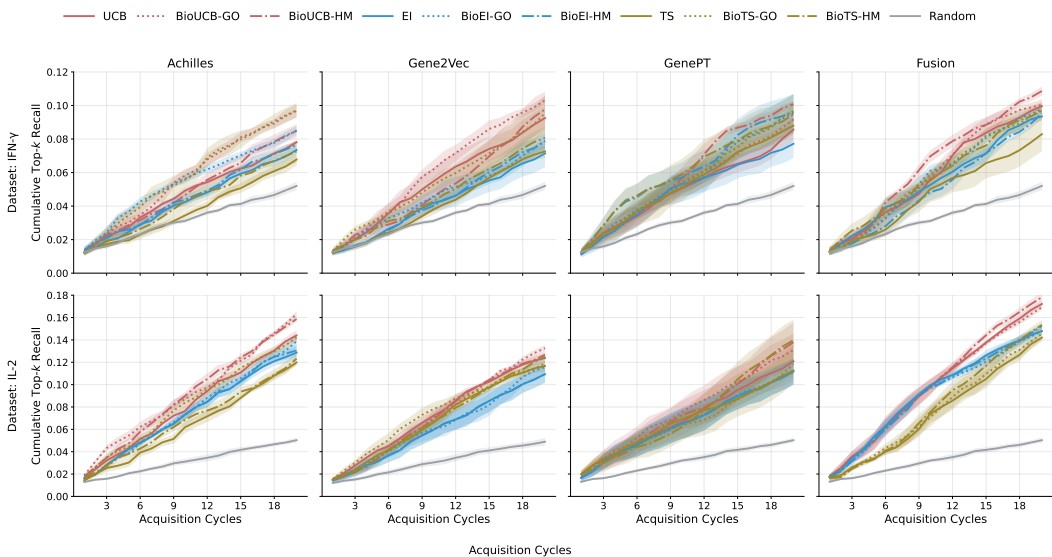

Figure 6: **Performance of standard BO and BioBO with three different modalities and their fusion for IFN-γ (top) and IL-2 (bottom)**. We observe that BO with Fusion is better than BO with any single modality and BioBO that incorporates priors from enrichment analysis is better than the corresponding BO without prior.

Figure 6 shows the complete BO results across both datasets and all four representations (Achilles, Gene2Vec, GenePT, Fusion), where biology-informed variants (BioUCB, BioEI, BioTS) with enrichment analysis significantly exceed their base counterparts (UCB, EI, TS), and surrogate models with fused gene embeddings are better than any single modality. Improvements are most evident in early–mid cycles (better sample efficiency) and narrow later as methods converge. UCB remains a strong base acquisition function, and the random baseline is consistently inferior.

## E.3 CORRELATIONS BETWEEN THE PERFORMANCE OF BO AND SURROGATE MODEL

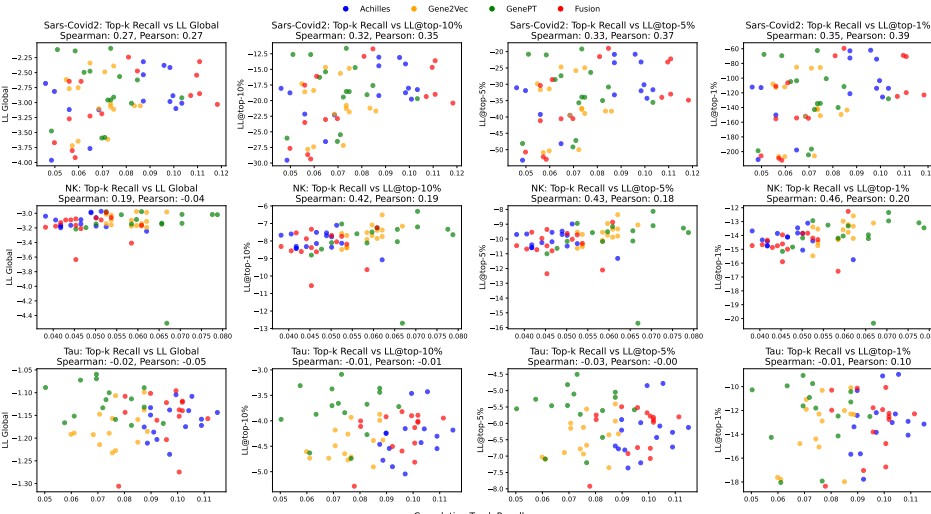

Figure 7: Relations between the performance of BO (measured by cumulative top-k recall) and surrogate model (measured by LL) on Tau, NK, and Sars-Covid2. We observed that the performance of BO is more correlated with the performance of the surrogate model near optimal (LL@top-1%) compared with global.

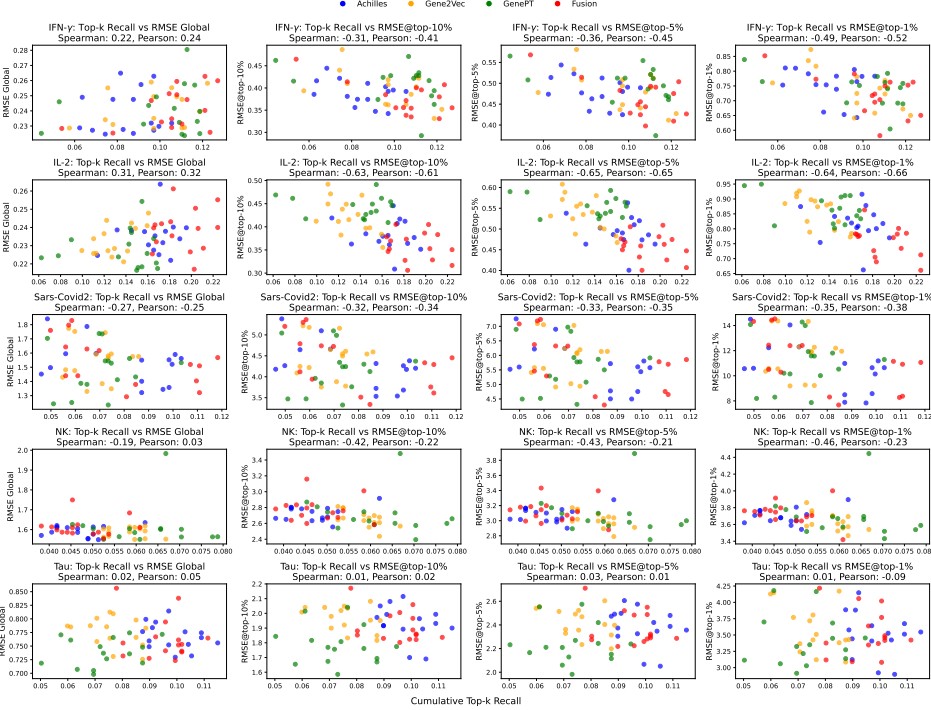

Figure 8: Relations between the performance of BO (measured by cumulative top-k recall) and surrogate model (measured by RMSE) on 5 datasets. We observed that the performance of BO is more correlated with the performance of the surrogate model near optimal (RMSE@top-1%) compared with global.

## E.4 PURE EA RESULTS FOR TAU, NK, AND SARS-COVID2

We show the performance of experimental designs using enrichment analysis only and using BioUCB for Tau, NK, and Sars-Covid2 datasets with Achilles on Figure 9. We observe that in most cases, pure EA is similar to random on all three datasets, except for EA with GO on Tau where BioUCB is better than UCB.

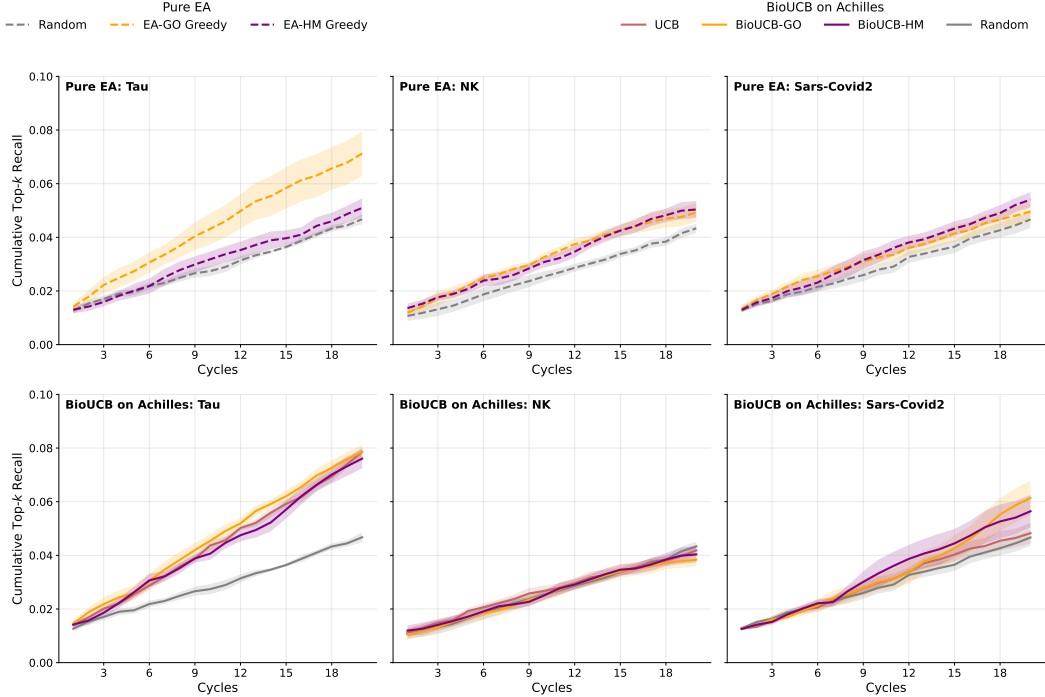

Figure 9: Performance of pure EA and BioUCB on Achilles for Tau, NK, and Sars-Covid2.

## E.5 BO RESULTS FOR TAU, NK, AND SARS-COVID2

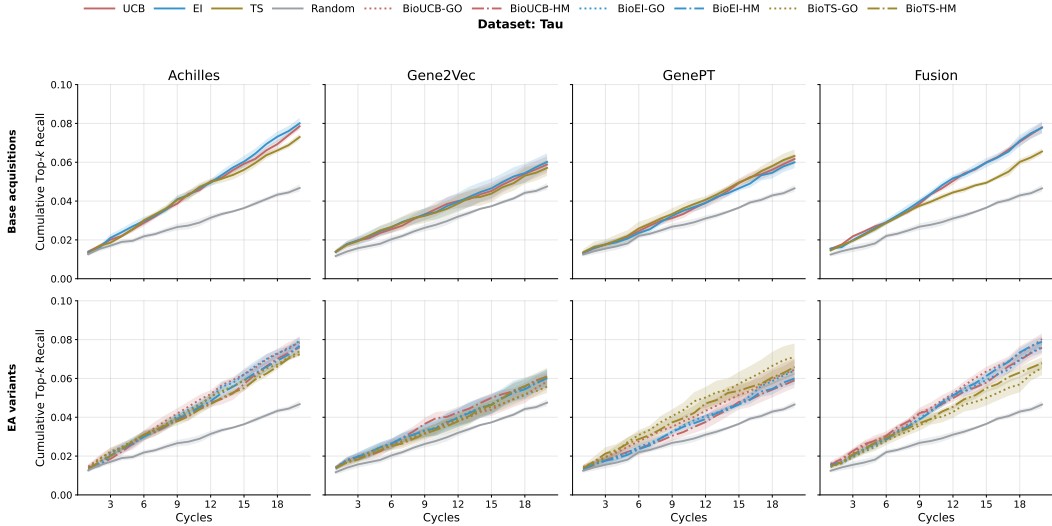

Figure 10: Performance of BO and BioBO with three modalities and their fusion for Tau.

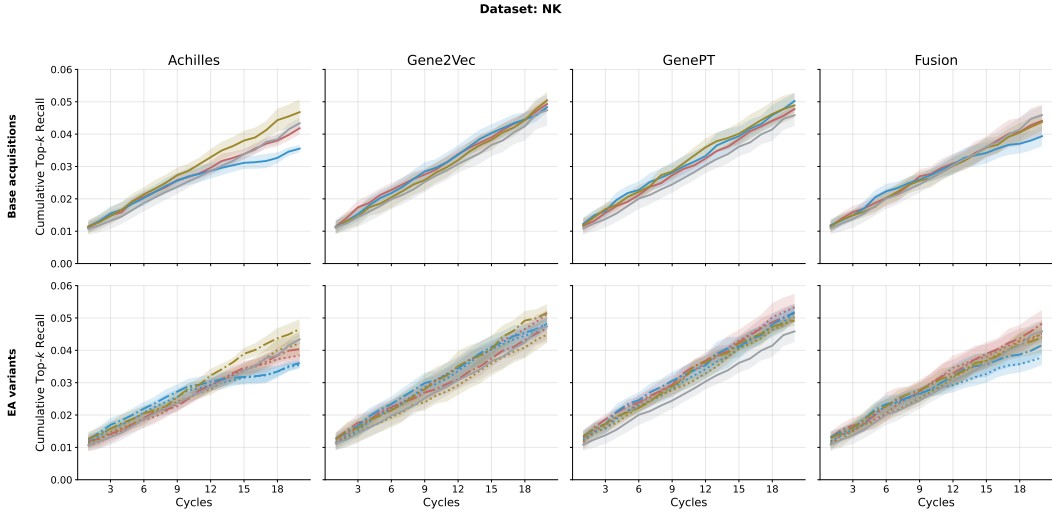

Figure 11: Performance of BO and BioBO with three modalities and their fusion for NK.

## F LATENT-SPACE FUSION FOR MULTIMODAL SURROGATE MODELS

To better integrate heterogeneous biological modalities in Bayesian Optimization (BO) surrogate models, we implement a **latent-space fusion** strategy. Each modality $x_1, x_2, x_3$ is first projected via modality-specific fully connected layers (with dropout for uncertainty), then fused in the latent space via either concatenation or cross-attention, followed by a final Bayesian MLP to predict the response:

$$y = \text{fc3}\big(\text{fc2}(\text{cross\_attention}(\text{fc11}(x_1), \text{fc12}(x_2), \text{fc13}(x_3))))$$

or

$$y = \text{fc3}\big(\text{fc2}(\text{concatenation}(\text{fc11}(x_1), \text{fc12}(x_2), \text{fc13}(x_3))))$$

This allows the surrogate to capture cross-modal interactions more effectively than simple concatenation of raw embeddings.

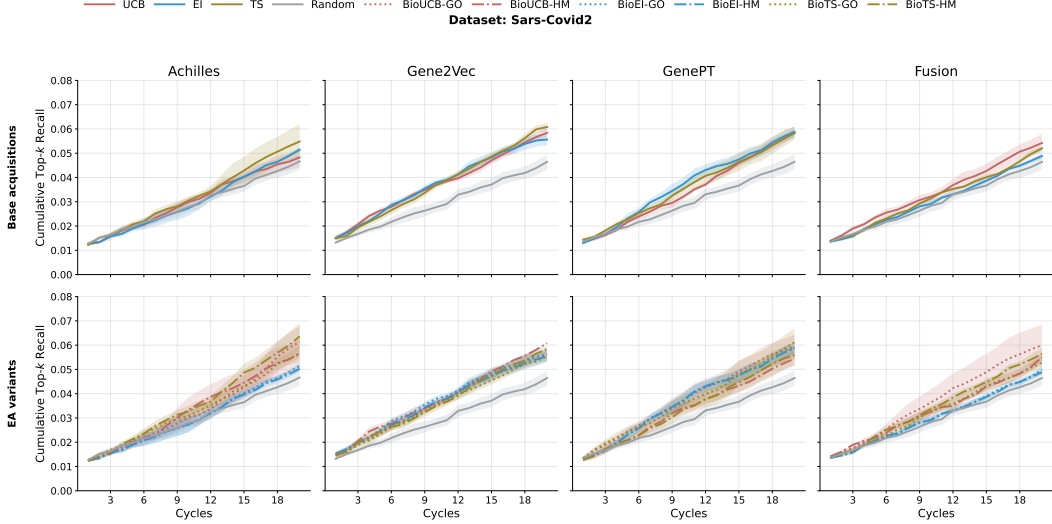

Figure 12: Performance of BO and BioBO with three modalities and their fusion for Sars-Covid2.

We evaluate three surrogate variants: standard Bayesian MLP (single-modal), latent concatenation, and latent attention. Results for two datasets, IFN-$\gamma$ and IL2, are shown in Table 3.

| Acquisition | IFN-$\gamma$ | | | IL2 | | |
|---|---|---|---|---|---|---|
| | Bayesian MLP | Latent Concatenation | Latent Attention | Bayesian MLP | Latent Concatenation | Latent Attention |
| EI | 0.093 (0.001) | 0.102 (0.001) | 0.109 (0.002) | 0.148 (0.002) | 0.141 (0.002) | 0.164 (0.002) |
| BioEI-GO | 0.098 (0.000) | 0.107 (0.001) | 0.115 (0.004) | 0.147 (0.003) | 0.143 (0.003) | 0.166 (0.003) |
| BioEI-HM | 0.096 (0.001) | 0.108 (0.001) | 0.116 (0.003) | 0.153 (0.002) | 0.154 (0.002) | 0.174 (0.002) |
| TS | 0.083 (0.001) | 0.099 (0.003) | 0.107 (0.001) | 0.142 (0.001) | 0.142 (0.002) | 0.166 (0.004) |
| BioTS-GO | 0.095 (0.001) | 0.105 (0.001) | 0.109 (0.002) | 0.147 (0.003) | 0.139 (0.001) | 0.160 (0.002) |
| BioTS-HM | 0.097 (0.001) | 0.110 (0.003) | 0.124 (0.002) | 0.153 (0.002) | 0.154 (0.002) | 0.175 (0.003) |
| UCB | 0.100 (0.001) | 0.102 (0.003) | 0.113 (0.000) | 0.174 (0.001) | 0.155 (0.001) | 0.169 (0.002) |
| BioUCB-GO | 0.102 (0.001) | 0.101 (0.004) | 0.116 (0.003) | 0.169 (0.001) | 0.173 (0.003) | 0.173 (0.002) |
| BioUCB-HM | 0.109 (0.001) | 0.112 (0.002) | 0.127 (0.003) | 0.178 (0.001) | 0.168 (0.002) | 0.176 (0.005) |

Table 3: Comparison of BO performance using different latent-space fusion strategies on IFN-$\gamma$ and IL2 datasets (mean ± std over 5 seeds).

Latent attention consistently outperforms latent concatenation and the Bayesian MLP across acquisition functions for both datasets, particularly for BioUCB-HM, where cross-modal interactions are critical. Latent concatenation also improves over the single-modal MLP, confirming the benefit of integrating multiple modalities. These results support the claim in the main text that multimodality fusion enhances BO efficiency.

## G RUNTIME COMPARISON

We report average runtime per iteration (evaluating 20 genes per cycle) for BioBO and baseline BO methods over all datasets. All experiments were run on a standard GPU (NVIDIA A10). The table includes variants with and without multimodal fusion and enrichment analysis (EA) to show the computational overhead introduced by these components. While multimodal fusion and EA slightly increase runtime compared to single-modality models, the additional cost remains modest and practical for typical high-throughput CRISPR experiments.

## H SENSITIVITY TO THE TOP-k% THRESHOLD IN ENRICHMENT ANALYSIS

We evaluate the effect of different top-k% thresholds used for enrichment analysis, varying k from 5% to 50% on the IFN-$\gamma$ dataset (Achilles features). As shown in Table 5, BioBO remains robust for k between 5–20%, exhibiting only minor performance variation. Larger thresholds (30–50%)

| Method | Avg Runtime per BO Cycle (s) |
|---|---|
| UCB (Achilles) | 8.55 |
| UCB (Fusion) | 10.50 |
| BioUCB-HM (Fusion) | 12.45 |
| EI (Achilles) | 7.64 |
| EI (Fusion) | 12.57 |
| BioEI-HM (Fusion) | 13.05 |
| TS (Achilles) | 6.95 |
| TS (Fusion) | 12.18 |
| BioTS-HM (Fusion) | 12.87 |

Table 4: Runtime per iteration for BioBO and baseline BO methods, averaged over datasets. Variants with multimodal fusion and/or enrichment analysis (EA) are included to show the overhead of these components.

dilute the enrichment signal by including a broader, noisier set of genes, leading to slightly reduced BO performance. We use k = 10% as a practical default.

| Top-k% | 5% | 10% | 15% | 20% | 30% | 50% |
|---|---|---|---|---|---|---|
| BioEI-GO | $0.090 \pm 0.001$ | $0.085 \pm 0.006$ | $0.084 \pm 0.009$ | $0.085 \pm 0.010$ | $0.074 \pm 0.002$ | $0.071 \pm 0.001$ |

Table 5: Sensitivity of BioBO to top-k% used for enrichment analysis. Performance shown as cumulative top-k recall.

# I  INTERPRETABILITY CASE STUDY ON IL-2 DATASET

To assess whether the interpretability benefits of BioBO generalize beyond the IFN-$\gamma$ setting, we analyze the IL-2 immune-cell CRISPR perturbation dataset. The results mirror the IFN-$\gamma$ findings: baseline UCB recovers several relevant pathways but with modest enrichment strength, whereas BioUCB-HM identifies the same pathways with higher overlap and stronger statistical significance.

Table 6: Enrichment analysis on IL-2 dataset comparing UCB and BioUCB-HM.

| Baseline UCB | | | | |
|---|---|---|---|---|
| Pathway | Overlap | Adjusted p-value | Odds Ratio | Combined Score |
| MYC_TARGETS_V1 | 49/200 | $6.16 \times 10^{-27}$ | 11.512 | 738.585 |
| E2F_TARGETS | 36/200 | $8.76 \times 10^{-15}$ | 7.004 | 248.528 |
| G2M_CHECKPOINT | 26/200 | $2.01 \times 10^{-7}$ | 4.436 | 80.423 |
| DNA_REPAIR | 22/150 | $2.41 \times 10^{-7}$ | 5.028 | 88.784 |
| BioUCB-HM (Fusion + EA) | | | | |
| MYC_TARGETS_V1 | 179/200 | $2.39 \times 10^{-231}$ | 487.252 | 260596 |
| E2F_TARGETS | 41/200 | $2.45 \times 10^{-12}$ | 4.962 | 148.099 |
| MYC_TARGETS_V2 | 18/58 | $1.79 \times 10^{-8}$ | 8.081 | 166.006 |
| G2M_CHECKPOINT | 32/200 | $5.50 \times 10^{-7}$ | 3.518 | 59.221 |

These results show that BioUCB-HM not only recovers all pathways identified by UCB but also enhances their enrichment signal by several orders of magnitude. Mechanistically, these pathways—MYC, E2F, G2M checkpoint, and DNA repair—govern central processes in lymphocyte metabolism and proliferation (Ren et al., 2002; DeGregori et al., 1997; Wang et al., 2011; Rathmell, 2011). The stronger enrichment observed under BioUCB-HM reflects its ability to prioritize perturbations that align with the regulatory circuitry of immune-cell activation, providing deeper mechanistic insight into pathway-level drivers. This analysis was also validated by two independent immunology domain experts.

## J  FAILURE CASES: MISMATCHED ENRICHMENT PATHWAYS

Although enrichment analysis substantially strengthens acquisition when the pathway database is relevant to the biological context, we also evaluate failure cases where the enrichment prior is mismatched. Specifically, we apply oncology-focused pathways ("ONC") to guide the design of immune-cell perturbations. Because these pathways are not related to immune signaling, the enrichment prior becomes non-informative and may slightly bias the acquisition towards irrelevant genes. In such settings, BioBO effectively falls back to the multimodal surrogate model, resulting in little or no improvement over the baseline BO acquisition.

Table 7: Failure-case comparison on IFN-$\gamma$: correct enrichment prior (GO) vs. mismatched oncology prior (ONC).

| Method | Fusion | Achilles | GPT | Gene2vec |
|---|---|---|---|---|
| EI | 0.093 (0.001) | 0.072 (0.001) | 0.077 (0.004) | 0.071 (0.006) |
| BioEI-GO (correct) | 0.098 (0.000) | 0.085 (0.000) | 0.095 (0.005) | 0.079 (0.004) |
| BioEI-ONC (mismatched) | 0.091 (0.001) | 0.074 (0.002) | 0.077 (0.008) | 0.073 (0.006) |

These results reinforce the practical takeaway: BioBO provides strong gains when pathway knowledge is biologically aligned with the experimental setting, while remaining robust when the prior is noisy or mismatched—consistent with our theoretical no-harm guarantee.

## K  ROBUSTNESS TO MISSING MODALITIES

To assess BioBO's robustness in practical scenarios where some embedding modalities are unavailable, we simulate missing data on the IFN-$\gamma$ dataset by dropping selected modalities and imputing missing embeddings with KNN. Table 8 summarizes the performance (mean $\pm$ std over 7 seeds) across three acquisition functions. Fusion remains consistently superior to single-modality surrogates even under KNN-imputation, indicating that heterogeneous embeddings provide complementary biological signal and that BioBO remains usable when embeddings are partially missing—a common situation in large-scale perturbation screens.

Table 8: Performance when some modalities are missing on IFN-$\gamma$ dataset. KNN-imputation is used for missing embeddings.

| Modality | EI | UCB | TS |
|---|---|---|---|
| Fusion (all modalities; 18,344 genes) | $0.046 \pm 0.003$ | $0.060 \pm 0.001$ | $0.048 \pm 0.002$ |
| Achilles only | $0.040 \pm 0.001$ | $0.042 \pm 0.002$ | $0.041 \pm 0.001$ |
| GPT only | $0.034 \pm 0.002$ | $0.050 \pm 0.008$ | $0.041 \pm 0.008$ |
| Gene2Vec only | $0.028 \pm 0.003$ | $0.035 \pm 0.003$ | $0.033 \pm 0.003$ |

These results show that multimodal fusion remains advantageous even under partially missing data, reflecting the complementary structure of gene-level biological embeddings and supporting the practical deployability of BioBO.

## L  INTERPRETATION OF ENRICHMENT PARAMETERS $t$ AND $\beta$

The temperature parameter $t$ and the prior strength $\beta$ control the contribution of enrichment analysis (EA) to the acquisition function relative to the uncertainty of the surrogate model. Conceptually, $t$ determines how concentrated the enrichment-derived prior is across candidate genes: as $t \to \infty$, the prior becomes uniform, and EA is ignored (pure exploration), whereas as $t \to 0$, the prior becomes sharply peaked, emphasizing top-ranked genes (heavy exploitation). The parameter $\beta$ modulates the weight of this prior within the acquisition: very small $\beta$ effectively removes the influence of EA, while extremely large $\beta$ over-amplifies the enrichment signals. Empirically, moderate values of

$t$ and $\beta$ provide stable performance, balancing exploitation of enriched pathways with exploration guided by the surrogate model. This discussion complements the main text and provides practical guidance for setting these hyperparameters.

## M    ENSEMBLING MULTIPLE ENRICHMENT SOURCES

BioBO is fully compatible with ensembling multiple enrichment sources because the $\pi$-BO prior formulation allows additive or multiplicative aggregation of priors. While the main text reports GO and Hallmark (HM) separately for clarity, we conducted experiments using an ensemble prior that averages enrichment-derived scores from both databases ("BioEI-GOHM"). Table 9 summarizes the performance across modalities on the IFN-$\gamma$ dataset.

Table 9: Performance of BioBO with multiple enrichment sources. BioEI-GOHM averages GO and Hallmark priors, showing consistent improvement over individual priors.

| Method | Fusion | Achilles | GPT | Gene2Vec |
|---|---|---|---|---|
| EI | 0.093 (0.001) | 0.072 (0.001) | 0.077 (0.004) | 0.071 (0.006) |
| BioEI-GO | 0.098 (0.000) | 0.085 (0.000) | 0.095 (0.005) | 0.079 (0.004) |
| BioEI-HM | 0.096 (0.001) | 0.076 (0.001) | 0.096 (0.007) | 0.079 (0.002) |
| BioEI-GOHM | 0.101 (0.002) | 0.092 (0.004) | 0.093 (0.008) | 0.084 (0.004) |

These results demonstrate that BioBO can naturally leverage complementary strengths from multiple enrichment sources, and ensemble priors consistently match or outperform individual priors. More dynamic weighting strategies for combining enrichment sources are a promising direction for future work.

## N    FURTHER INTERPRETABILITY ANALYSIS OF UNDEREXPLORED BIOLOGICALLY NOVEL GENES PRIORITIZED BY BIOBO

Beyond the well-known MYC/E2F modules reported in the main text, BioBO prioritized a set of underexplored genes whose knockouts produced top 0.1% IFN-$\gamma$ increases. These include FAU, MAK16, PCBP2, and multiple ribosomal proteins (e.g., RPL19, RPL27, RPL37, RPS11, RPS13, RPS17, RPS20). These genes are not typically highlighted by baseline BO, yet they form a coherent module downstream of MYC-driven ribosome biogenesis, a key regulator of T-cell growth and effector differentiation (Destefanis et al., 2020). Perturbation of ribosomal components induces nucleolar stress and NF-$\kappa$B/p53 activation (Akef et al., 2020), shifting cells from proliferation toward higher cytokine output. PCBP2 further modulates MAVS/RIG-I signaling Onomoto et al. (2021), linking directly to interferon pathways.

Two independent domain experts reviewed and validated this mechanistic interpretation. This analysis provides concrete examples of how BioBO's enrichment-informed acquisition can reveal biologically meaningful, underexplored targets, complementing standard BO approaches.

## O    ADDITIONAL RELATIONAL ANALYSIS BETWEEN BO PERFORMANCE AND SURROGATE MODELING

We measure global LL across all genes and observe a weak or even negative correlation with BO performance. This arises because global LL is dominated by the dense region of low-response genes, whereas BO only depends on the surrogate in a small neighborhood of the maximizer. In our CRISPR datasets, only a small fraction of genes have a high response to IFN-$\gamma$ / IL-2. A surrogate that fits the bulk region extremely well (high global LL) but underestimates the tails can perform worse in BO than one that slightly sacrifices global LL but better resolves the local geometry near the optimum. When we restrict LL to the top-k genes (in terms of ground truth response), the correlation with BO performance becomes positive and substantially stronger (see Appendix X), confirming that local surrogate quality near the optimum, rather than global goodness-of-fit, is what drives BO.

Table 10: Conditional correlation analysis for IFN-$\gamma$ across fusion strategies and acquisition functions.

| Fusion | Method | LL Global | LL@10% | LL@5% | LL@1% |
|---|---|---|---|---|---|
| None | EI | 0.051 | 0.211 | 0.202 | 0.212 |
| | TS | 0.038 | 0.038 | 0.059 | 0.153 |
| | UCB | -0.065 | 0.285 | 0.315 | 0.356 |
| | Random | 0.181 | 0.008 | -0.009 | -0.035 |
| Input concat. | EI | -0.018 | 0.313 | 0.292 | 0.298 |
| | TS | 0.096 | 0.163 | 0.169 | 0.198 |
| | UCB | -0.054 | 0.362 | 0.357 | 0.355 |
| | Random | 0.209 | 0.037 | 0.003 | -0.037 |
| Latent concat. | EI | 0.005 | 0.246 | 0.244 | 0.282 |
| | TS | 0.078 | 0.140 | 0.145 | 0.205 |
| | UCB | -0.067 | 0.273 | 0.299 | 0.364 |
| | Random | 0.181 | 0.008 | -0.009 | -0.035 |
| Latent attention | EI | -0.021 | 0.216 | 0.219 | 0.262 |
| | TS | 0.070 | 0.151 | 0.178 | 0.270 |
| | UCB | -0.135 | 0.285 | 0.314 | 0.382 |
| | Random | 0.181 | 0.008 | -0.009 | -0.035 |

Table 11: Conditional correlation analysis for IL-2 across fusion strategies and acquisition functions.

| Fusion | Method | LL Global | LL@10% | LL@5% | LL@1% |
|---|---|---|---|---|---|
| None | EI | -0.152 | 0.455 | 0.471 | 0.479 |
| | TS | 0.007 | 0.325 | 0.362 | 0.386 |
| | UCB | -0.139 | 0.493 | 0.515 | 0.515 |
| | Random | 0.088 | 0.224 | 0.233 | 0.208 |
| Input concat. | EI | -0.173 | 0.572 | 0.581 | 0.568 |
| | TS | -0.023 | 0.439 | 0.457 | 0.441 |
| | UCB | -0.217 | 0.540 | 0.559 | 0.546 |
| | Random | 0.088 | 0.274 | 0.278 | 0.248 |
| Latent concat. | EI | -0.197 | 0.524 | 0.542 | 0.526 |
| | TS | -0.026 | 0.375 | 0.404 | 0.402 |
| | UCB | -0.192 | 0.479 | 0.506 | 0.500 |
| | Random | 0.088 | 0.224 | 0.233 | 0.208 |
| Latent attention | EI | -0.153 | 0.552 | 0.568 | 0.559 |
| | TS | -0.058 | 0.419 | 0.449 | 0.451 |
| | UCB | -0.171 | 0.514 | 0.532 | 0.530 |
| | Random | 0.088 | 0.224 | 0.233 | 0.208 |

# P ADJUSTED CUMULATIVE TOP-K RECALL WITH $\pm 1.96$ S.E.M.

Table 12: **Cumulative top-k recall with 1.96 s.e.m. of each acquisition function on different datasets**. We observe that BioBO achieves the best performance on 23/24 different settings, and BioUCB-HM with surrogate function using fused features achieves the best performance for both IFN-$\gamma$ and IL-2. The best performance (with the smallest standard error) is bold.

| Phenotype: IFN-$\gamma$ | Fusion | Achilles | GenePT | Gene2Vec |
|---|---|---|---|---|
| EI | 0.093 (0.002) | 0.072 (0.002) | 0.077 (0.008) | 0.071 (0.011) |
| BioEI-GO (ours) | **0.098 (0.001)** | **0.085 (0.001)** | 0.095 (0.010) | 0.079 (0.008) |
| BioEI-HM (ours) | 0.096 (0.002) | 0.076 (0.002) | **0.096 (0.014)** | **0.079 (0.004)** |
| TS | 0.083 (0.002) | 0.068 (0.002) | 0.088 (0.004) | 0.073 (0.004) |
| BioTS-GO (ours) | 0.095 (0.002) | 0.073 (0.001) | **0.097 (0.008)** | **0.095 (0.009)** |
| BioTS-HM (ours) | **0.097 (0.002)** | **0.097 (0.009)** | 0.093 (0.010) | 0.081 (0.008) |
| UCB | 0.100 (0.002) | 0.077 (0.002) | 0.086 (0.008) | 0.093 (0.010) |
| BioUCB-GO (ours) | 0.102 (0.002) | **0.098 (0.004)** | 0.092 (0.010) | 0.098 (0.004) |
| BioUCB-HM (ours) | **0.109 (0.002)** | 0.085 (0.006) | **0.101 (0.002)** | **0.103 (0.008)** |
| Random | 0.050 (0.002) | 0.050 (0.002) | 0.050 (0.002) | 0.050 (0.002) |
| Phenotype: IL-2 | Fusion | Achilles | GenePT | Gene2Vec |
| EI | 0.148 (0.004) | 0.130 (0.006) | 0.107 (0.010) | 0.109 (0.004) |
| BioEI-GO (ours) | 0.147 (0.006) | **0.138 (0.006)** | 0.107 (0.010) | **0.115 (0.004)** |
| BioEI-HM (ours) | **0.153 (0.003)** | 0.130 (0.005) | 0.107 (0.009) | 0.109 (0.004) |
| TS | 0.142 (0.002) | 0.119 (0.002) | 0.113 (0.027) | 0.113 (0.004) |
| BioTS-GO (ours) | 0.147 (0.005) | **0.142 (0.004)** | 0.119 (0.021) | 0.119 (0.002) |
| BioTS-HM (ours) | **0.153 (0.004)** | 0.123 (0.007) | **0.139 (0.025)** | **0.124 (0.004)** |
| UCB | 0.174 (0.002) | 0.143 (0.006) | 0.118 (0.022) | 0.123 (0.001) |
| BioUCB-GO (ours) | 0.169 (0.002) | 0.158 (0.002) | 0.131 (0.015) | **0.133 (0.004)** |
| BioUCB-HM (ours) | **0.178 (0.002)** | **0.163 (0.002)** | **0.138 (0.023)** | 0.127 (0.001) |
| Random | 0.049 (0.002) | 0.048 (0.002) | 0.049 (0.002) | 0.046 (0.003) |

