# OpenReview forum: "BioBO: Biology-informed Bayesian Optimization for Perturbation Design"
_ICLR.cc/2026/Conference — ICLR 2026 Poster_

### Official Review · Reviewer_ohwq · 2025-10-19

**Soundness:** 3
**Presentation:** 3
**Contribution:** 4
**Rating:** 8
**Confidence:** 4

**Summary:**

This paper proposes _Biology-Informed Bayesian Optimization (BioBO)_, which addresses challenges in drug discovery that require genomic intervention experiments. The authors introduce two key innovations: (1) a multimodal gene representation to improve the surrogate model in Bayesian Optimization (BO), and (2) the integration of enrichment analysis (EA) into a biology-prior–augmented acquisition function through a principled πBO framework. This integration provides a theoretical “no-worse” guarantee and establishes a foundation for incorporating broader sources of biological knowledge in similar ways.

Experiments on real-world problems demonstrate the effectiveness of both proposed components — the multimodal embeddings and the biological priors. The results produced by BioBO are also biologically interpretable.

**Strengths:**

- The paper addresses a well-motivated and highly important problem.

- It integrates EA-based experimental design common in biology into the πBO framework,  bridging knowledge from two domains, and offers a principled non-worse theoretical guarantee.

- The ablation studies sufficiently demonstrate the contributions of the multimodal representations and the EA integration. **Furthermore, the genes selected by BioBO align with known biological interpretations of central regulatory genes, indicating strong real-world potential.**

- The multimodal surrogate model is modular, allowing a wide range of biological and machine learning encoders, which enhances its practicality for real-world applications.

**Weaknesses:**

- According to the appendix, the width of the error bars is ±1 s.e.m., which is less convincing compared to the standard ±1.96 s.e.m. used for 5% significance. Could the authors justify this choice?

- _(Minor)_ The generality of the method is limited. Although the proposed approach is termed “Biology-Informed BO,” it applies primarily to biological problems where EA is feasible. The authors might consider adjusting the algorithm’s name to better reflect its domain specificity.

**Questions:**

1. The multimodal design appears to involve simple concatenation of multiple gene embeddings. Is there a specific reason for choosing this approach to combine different embeddings?

2. There is little discussion of the individual contributions of each gene embedding or the rationale for selecting each. Could the authors provide more insight into how one should choose among available gene embeddings?

3. If I understand correctly, the so-called “biological prior” is derived from statistical patterns in the training data, rather than from external biological knowledge. Could the authors clarify how EA can still enhance BO performance under this setup, even without access to external priors?

---

> ### Author Response · Authors · 2025-11-19
> **Initial Point by Point Rebuttal (and invitation for further discussion)- Response 1 of 2**
>
> We thank the reviewer for recognizing the strong motivation of our work, the principled integration of enrichment analysis into the π-BO framework with a no-worse theoretical guarantee, the clear ablation studies demonstrating the contributions of multimodal embeddings and EA, and the practical modularity and biological interpretability of our approach. Below we address the questions and comments of the reviewer point-by-point.
>
> **1. Exploration of fusion mechanism beyond concatenation in input space:** We thank the reviewer for suggesting more advanced fusion approaches. We have now implemented latent-space fusion with the three modalities. Each modality \(x_1, x_2, x_3\) is first projected via modality-specific fully connected layers (with dropout for uncertainty), then fused either via concatenation (called latent concatenation) or cross-attention (called latent attention) in the latent space, followed by a final Bayesian MLP.
>
> We report the results for this for  IFN-γ dataset (mean ± std over 7 seeds) below and add this along with results for IL-2 dataset in Appendix F and referred to in main text in lines 193-194. Latent attention consistently outperforms latent concatenation and the Bayesian MLP concatentaion in input space (original fusion strategy in paper) across acquisition functions for both datasets, particularly for BioUCB-HM. This confirms the benefit of integrating multiple modalities and reinforcing our prior results even further.
>
> | Model | EI | BioEI-GO | BioEI-HM | TS | BioTS-GO | BioTS-HM | UCB | BioUCB-GO | BioUCB-HM |
> |-------|----|-----------|----------|----|-----------|----------|-----|------------|-----------|
> | Bayesian MLP | 0.093 ± 0.001 | 0.098 ± 0.000 | 0.096 ± 0.001 | 0.083 ± 0.001 | 0.095 ± 0.001 | 0.097 ± 0.001 | 0.100 ± 0.001 | 0.102 ± 0.001 | 0.109 ± 0.001 |
> | Latent Concatenation | 0.102 ± 0.001 | 0.107 ± 0.000 | 0.108 ± 0.001 | 0.099 ± 0.003 | 0.105 ± 0.001 | 0.110 ± 0.003 | 0.102 ± 0.003 | 0.101 ± 0.004 | 0.112 ± 0.002 |
> | Latent Attention | 0.109 ± 0.002 | 0.115 ± 0.004 | 0.116 ± 0.003 | 0.107 ± 0.001 | 0.109 ± 0.002 | 0.124 ± 0.002 | 0.113 ± 0.000 | 0.116 ± 0.003 | 0.127 ± 0.003 |
>
> **2. How should one choose among available gene embeddings?:** Practitioners can choose among available gene embeddings based on the types of prior knowledge they trust and the modalities available in their setting. We can also use the predictive performance of the surrogate model as an indicator to select modality. If a model (e.g., MLP trained on STRING) is similar to a mean predictor, we don't need to add this modality. This won't rely on running another BO for hyper-parameter selection which might be tricky in real-world applications.  In our experiments, different embeddings we used capture complementary biological signals: Achilles represents gene-knockout phenotypes, Gene2Vec encodes co-expression and network topology, and GPT-based embeddings capture sequence- and function-level semantics.
>
> **3. How EA can still enhance BO performance under this setup, even without access to external priors?:** Enrichment analysis (EA) provides pathway-level information, capturing how perturbing one gene may affect the phenotypes of others and which genes are functionally related. Each pathway is purely external information/knowledge, independent of the gene selection pool for perturbation design. By identifying statistically enriched pathways linked with the top predicted genes from the surrogate model, EA can bias experimental design toward coherent gene modules, improving the exploitation of high-potential perturbations. This explains why EA alone can more efficiently select impactful genes than random acquisition. However, EA by itself lacks principled exploration and may overlook under-characterized or novel genes. BioBO addresses this limitation by integrating EA within the π-BO framework, maintaining a balanced exploration–exploitation trade-off while leveraging pathway-level structure to guide selection toward biologically meaningful and high-performing perturbations.

---

> > ### Author Response · Authors · 2025-11-19
> > **Initial Point by Point Rebuttal (and invitation for further discussion)- Response 2 of 2**
> >
> > **4. Width of the error bars is ±1 s.e.m., which is less convincing compared to the standard ±1.96 s.e.m. used for 5% significance:** We reported ±1 s.e.m. for clarity and comparability with prior BO studies. Using ±1.96 s.e.m. (≈95% CI) would increase the bar width but does not change the qualitative conclusions. For completeness, we have provided a full 95% CI for main BO results in the Appendix P and also report them below.
> >
> > ### Phenotype: IFN-γ
> >
> > | Method            | Fusion          | Achilles        | GenePT          | Gene2Vec        |
> > |-------------------|-----------------|-----------------|------------------|------------------|
> > | EI                | 0.093 (0.002)   | 0.072 (0.002)   | 0.077 (0.008)    | 0.071 (0.011)    |
> > | BioEI-GO (ours)   | 0.098 (0.001)   | 0.085 (0.001)   | 0.095 (0.010)    | 0.079 (0.008)    |
> > | BioEI-HM (ours)   | 0.096 (0.002)   | 0.076 (0.002)   | 0.096 (0.014)    | 0.079 (0.004)    |
> > | TS                | 0.083 (0.002)   | 0.068 (0.002)   | 0.088 (0.004)    | 0.073 (0.004)    |
> > | BioTS-GO (ours)   | 0.095 (0.002)   | 0.073 (0.001)   | 0.097 (0.008)    | 0.095 (0.009)    |
> > | BioTS-HM (ours)   | 0.097 (0.002)   | 0.097 (0.009)   | 0.093 (0.010)    | 0.081 (0.008)    |
> > | UCB               | 0.100 (0.002)   | 0.077 (0.002)   | 0.086 (0.008)    | 0.093 (0.010)    |
> > | BioUCB-GO (ours)  | 0.102 (0.002)   | 0.098 (0.004)   | 0.092 (0.010)    | 0.098 (0.004)    |
> > | BioUCB-HM (ours)  | 0.109 (0.002)   | 0.085 (0.006)   | 0.101 (0.002)    | 0.103 (0.008)    |
> > | Random            | 0.050 (0.002)   | 0.050 (0.002)   | 0.050 (0.002)    | 0.050 (0.002)    |
> >
> >
> > ### Phenotype: IL-2
> >
> > | Method            | Fusion          | Achilles        | GenePT          | Gene2Vec        |
> > |-------------------|-----------------|-----------------|------------------|------------------|
> > | EI                | 0.148 (0.004)   | 0.130 (0.006)   | 0.107 (0.010)    | 0.109 (0.004)    |
> > | BioEI-GO (ours)   | 0.147 (0.006)   | 0.138 (0.006)   | 0.107 (0.010)    | 0.115 (0.004)    |
> > | BioEI-HM (ours)   | 0.153 (0.003)   | 0.130 (0.005)   | 0.107 (0.009)    | 0.109 (0.004)    |
> > | TS                | 0.142 (0.002)   | 0.119 (0.002)   | 0.113 (0.027)    | 0.113 (0.004)    |
> > | BioTS-GO (ours)   | 0.147 (0.005)   | 0.142 (0.004)   | 0.119 (0.021)    | 0.119 (0.002)    |
> > | BioTS-HM (ours)   | 0.153 (0.004)   | 0.123 (0.007)   | 0.139 (0.025)    | 0.124 (0.004)    |
> > | UCB               | 0.174 (0.002)   | 0.143 (0.006)   | 0.118 (0.022)    | 0.123 (0.001)    |
> > | BioUCB-GO (ours)  | 0.169 (0.002)   | 0.158 (0.002)   | 0.131 (0.015)    | 0.133 (0.004)    |
> > | BioUCB-HM (ours)  | 0.178 (0.002)   | 0.163 (0.002)   | 0.138 (0.023)    | 0.127 (0.001)    |
> > | Random            | 0.049 (0.002)   | 0.048 (0.002)   | 0.049 (0.002)    | 0.046 (0.003)    |
> >
> > **5. The proposed approach applies primarily to biological problems where EA is feasible:** We acknowledge that the method is specific to perturbation design where EA is applicable which is the focus application area of this paper.
> >
> > We hope our responses have addressed all your questions. We sincerely thank you again for your insightful comments and positive assessment of our work, which we greatly appreciate!

---

> > > ### Comment · Reviewer_ohwq · 2025-11-25
> > > **response**
> > >
> > > I appreciate the authors' effort to address all my issues, especially clarifying my misunderstanding about the external prior knowledge!

---

> > > > ### Author Response · Authors · 2025-11-28
> > > > **Thank you for your positive evaluation!**
> > > >
> > > > We thank the reviewer for their positive evaluation and suggestions. We are glad that our responses helped address all the issues!

---

### Official Review · Reviewer_cXha · 2025-10-29

**Soundness:** 3
**Presentation:** 3
**Contribution:** 3
**Rating:** 4
**Confidence:** 3

**Summary:**

The paper proposes BioBO* a biology-informed Bayesian optimization framework for efficient design of genomic perturbation experiments. The goal is to prioritize which gene knockouts to test in costly CRISPR assays. BioBO integrates two main ideas:
1. **Multimodal gene embeddings (Fusion)** – combining Achilles (functional dependency), Gene2Vec (co-expression), and GenePT (text-derived) representations to improve surrogate modeling.
2. **Enrichment analysis (EA)** – incorporating biologically meaningful pathway priors into acquisition functions via the π-BO framework (Hvarfner et al., 2022), guiding the search toward biologically coherent gene clusters.

The method is evaluated on five CRISPR datasets from GeneDisco, showing 25–40% labeling efficiency improvement over standard BO, and stronger enrichment signals in discovered pathways. The authors also provide theoretical guarantees that integrating EA priors does not worsen regret asymptotically.

**Strengths:**

- **Strong biological motivation**: Addresses a key challenge in experimental design for genomics, where exhaustive perturbation is infeasible.
- **Principled integration of biology and machine learning**: The use of the π-BO formalism to incorporate enrichment priors is theoretically grounded and non-trivial.
- **Novel multimodal surrogate modeling**: Demonstrates empirically that fusing heterogeneous biological embeddings (Achilles + Gene2Vec + GenePT) yields better performance near optima.
- **Comprehensive experiments**: The evaluation on multiple datasets (IFN-γ, IL-2, Tau, NK, SARS-CoV-2) with ablations and sensitivity analyses is thorough and carefully executed.
- **Interpretability and biological validation**: The enrichment results (e.g., MYC and E2F target pathways) provide mechanistic insight into immune regulation and show that the discovered perturbations are biologically meaningful.
- **No-harm theoretical guarantee**: Retains the asymptotic regret bounds of standard BO, ensuring robustness to potentially noisy enrichment priors.

**Weaknesses:**

- **Limited novelty in BO theory**: The core algorithmic novelty is the integration of known biological priors within an existing π-BO framework; the BO component itself remains myopic and standard.
- **Empirical improvements modest in some datasets**: While fusion provides consistent gains, improvements are smaller or marginal in certain conditions (e.g., BioEI on IL-2), suggesting dataset-dependent benefits.
- **Dependence on predefined pathway databases**: The approach relies on GO and Hallmark annotations; biases or incompleteness in these resources may limit generalization, especially in less-studied cell types.
- **Simple fusion strategy**: Concatenation is effective but basic. More principled fusion (e.g., learned or attention-based integration) could enhance representation quality.
- **No analysis of computational cost**: Training BNN surrogates with multimodal embeddings and enrichment updates may be computationally heavier than standard BO, but runtime comparisons are not reported.

**Questions:**

1. **Prior calibration**: How sensitive are the results to the choice of temperature parameter *t* and β in Eq. 4 and Eq. 6? The appendix shows some analysis, but can the authors comment on interpretability of these hyperparameters across datasets?
2. **Fusion mechanism**: Have the authors considered learning the fusion weights (e.g., via gating or attention) instead of static concatenation? Would that risk overfitting given small training sets per iteration?
3. **Pathway dependence**: Since GO and Hallmark differ in coverage and granularity, could BioBO dynamically switch or ensemble multiple enrichment sources?
4. **Scalability**: How would BioBO scale to genome-scale design (>20k genes) when multiple priors and embeddings are used? Are there computational bottlenecks?
5. **Biological interpretability**: Can the authors provide further examples where enrichment priors led to selection of underexplored but biologically novel genes, beyond MYC/E2F modules?

---

> ### Author Response · Authors · 2025-11-19
> **Initial Point by Point Rebuttal (and invitation for further discussion)- Response 1 of 3**
>
> We thank the reviewer for highlighting the strong biological motivation, principled integration of enrichment priors via π-BO, the novelty of multimodal surrogate modeling, comprehensive experiments, interpretability, and the no-harm theoretical guarantees of our work. Below we address all questions first and then address any remaining comments raised in Weaknesses section.
>
> **1. Interpretation of temperature parameter t and β:** We thank the reviewer for raising this question. Both the temperature parameter t (used to scale enrichment-derived priors) and β (controlling the strength of the prior within the acquisition) modulate the contribution of biological information relative to the surrogate’s uncertainty. Conceptually, t determines how “sharp” or “flat” the enrichment scores are, and therefore how concentrated the biological prior becomes across candidate genes. As t approaches infinity, enrichment based prior approaches uniform distribution and EA is ignored (exploration only) while if t approaches 0,  enrichment based prior approaches dirac-delta distribution with EA being heavily weighted (exploitation only). β controls the extent to which this prior influences the acquisition score. In Appendix C, we observe that very small β effectively removes the influence of biological prior (approaching exploration), whereas extremely large β over-amplifies enrichment signals (approaching exploitation). In practice, moderate values are stable for β. We have now also added this discussion in Appendix L.
>
> **2. Exploration of fusion mechanism beyond concatenation in input space:** We thank the reviewer for suggesting more advanced fusion approaches. We have now implemented latent-space fusion with the three modalities. Each modality \(x_1, x_2, x_3\) is first projected via modality-specific fully connected layers (with dropout for uncertainty), then fused either via concatenation (called latent concatenation) or cross-attention (called latent attention) in the latent space, followed by a final Bayesian MLP.
>
> We report the results for this for  IFN-γ dataset (mean ± std over 7 seeds) below and add this along with results for IL-2 dataset in Appendix F and referred to in main text in lines 193-194. Latent attention consistently outperforms latent concatenation and the Bayesian MLP concatentaion in input space (original fusion strategy in paper) across acquisition functions for both datasets, particularly for BioUCB-HM. This confirms the benefit of integrating multiple modalities and reinforcing our prior results even further.
> | Model | EI | BioEI-GO | BioEI-HM | TS | BioTS-GO | BioTS-HM | UCB | BioUCB-GO | BioUCB-HM |
> |-------|----|-----------|----------|----|-----------|----------|-----|------------|-----------|
> | Bayesian MLP | 0.093 ± 0.001 | 0.098 ± 0.000 | 0.096 ± 0.001 | 0.083 ± 0.001 | 0.095 ± 0.001 | 0.097 ± 0.001 | 0.100 ± 0.001 | 0.102 ± 0.001 | 0.109 ± 0.001 |
> | Latent Concatenation | 0.102 ± 0.001 | 0.107 ± 0.000 | 0.108 ± 0.001 | 0.099 ± 0.003 | 0.105 ± 0.001 | 0.110 ± 0.003 | 0.102 ± 0.003 | 0.101 ± 0.004 | 0.112 ± 0.002 |
> | Latent Attention | 0.109 ± 0.002 | 0.115 ± 0.004 | 0.116 ± 0.003 | 0.107 ± 0.001 | 0.109 ± 0.002 | 0.124 ± 0.002 | 0.113 ± 0.000 | 0.116 ± 0.003 | 0.127 ± 0.003 |
>
> **3. Ensembling multiple enrichment sources:** We thank the reviewer for this helpful suggestion. GO and Hallmark indeed differ in coverage and granularity, and BioBO is fully compatible with ensembling these sources. While the main paper reports GO and Hallmark separately for clarity, we have now conducted additional experiments using an ensemble prior that takes the union of enrichment-derived pathways from both GO and Hallmark databases (“BioEI-GOHM”). The results are shown in the table below and have been added to Appendix M and referred to in main text in lines 520-522. We find that the ensemble consistently matches or outperforms the individual GO or Hallmark derived priors across modalities. For instance, on the IFN-γ dataset, BioEI-GOHM achieves cumulative top-k recall of 0.101 (Fusion), 0.092 (Achilles), 0.093 (GPT), and 0.084 (Gene2Vec), improving upon or equaling BioEI-GO and BioEI-HM. This demonstrates that BioBO naturally accommodates multiple enrichment sources and can leverage their complementary strengths. More dynamic weighting strategies for enrichment sources are promising directions of future research.
>
> | Method | Fusion | Achilles | GPT | Gene2vec |
> |--------------|----------------|----------------|----------------|----------------|
> | EI | 0.093 (0.001) | 0.072 (0.001) | 0.077 (0.004) | 0.071 (0.006) |
> | BioEI-GO | 0.098 (0.000) | 0.085 (0.000) | 0.095 (0.005) | 0.079 (0.004) |
> | BioEI-HM | 0.096 (0.001) | 0.076 (0.001) | 0.096 (0.007) | 0.079 (0.002) |
> | BioEI-GOHM | 0.101 (0.002) | 0.092 (0.004) | 0.093 (0.008) | 0.084 (0.004) |

---

> ### Author Response · Authors · 2025-11-19
> **Initial Point by Point Rebuttal (and invitation for further discussion)- Response 2 of 3**
>
> **4. Scaling of BioBO/Computational cost:** We thank the reviewer for raising this important point. BioBO is designed to scale efficiently to genome-wide settings (>20k genes). In practice, the computational bottlenecks are the same as in standard BO: surrogate training (BNN) and acquisition maximization dominate runtime.  The acquisition maximization in BioBO scales linearly with the number of candidates. The run-time of surrogate model depends on embedding size rather than genome size, and training cost scales with the number of observed labels, not the size of the search space. Also, enrichment-based priors operate in linear time with respect to the number of genes.
>
> Runtime per iteration of our BioBO  is suitable for practical purposes (we have now included a runtime table for clarity in Appendix G and referred to it in Section 4.6 in main text). We also report it below. The table includes variants with and without multimodal fusion and enrichment analysis (EA) to show the computational overhead introduced by these two contributions of ours. While multimodal fusion and EA slightly increase runtime compared to single-modality (Achilles) models with different acquisition functions, the additional cost remains modest and practical for typical high-throughput CRISPR experiments.
> | Method | Average Runtime per BO Cycle (s) |
> |--------|---------|
> | UCB (Achilles) | 8.55 |
> | UCB (Fusion) | 10.50 |
> | BioUCB-HM (Fusion) | 12.45 |
> | EI (Achilles) | 7.64 |
> | EI (Fusion) | 12.57 |
> | BioEI-HM (Fusion) | 13.05 |
> | TS (Achilles) | 6.95 |
> | TS (Fusion) | 12.18 |
> | BioTS-HM (Fusion) | 12.87 |
>
> **5. More examples where enrichment priors led to selection of underexplored but biologically novel genes:** Beyond the MYC/E2F modules, BioBO also prioritized a set of underexplored genes—FAU, MAK16, PCBP2, and multiple ribosomal proteins (e.g., RPL19/27/37, RPS11/13/17/20)—whose knockouts produced top 0.1% IFN-γ increases. These genes are not typically highlighted by baseline BO but form a coherent module downstream of MYC-driven ribosome biogenesis, a well-established regulator of T-cell growth and effector differentiation [1]. Perturbing ribosomal components is known to induce nucleolar stress and NF-κB/p53 activation [2], shifting cells away from proliferation and toward higher cytokine output. PCBP2 additionally modulates MAVS/RIG-I signaling [3], providing a direct link to interferon pathways. Two independent domain experts also reviewed and validated this mechanistic interpretation. We have now added this analysis, including references and gene lists, to Appendix N and referred to it in main text in lines 463-464.
>
> **Responses to some of the Weaknesses not covered in Questions are presented below**
>
> **1. Dependence on predefined pathway databases:** We thank the reviewer for highlighting this. BioBO’s enrichment-based prior is indeed dependent on the relevance of the pathway database to the biological context. When the database matches the experimental system (e.g., GO for immune-cell phenotypes), BioBO consistently improves over EI—as shown by BioEI-GO outperforming EI across all modalities. However, when we intentionally substitute an irrelevant database (e.g., oncology-specific pathways, BioEI-ONC)  to immune-cell experiments, the gains diminish and performance nearly reverts to the baseline EI results. This demonstrates that BioBO degrades gracefully: in the worst case, it behaves similarly to standard BO, and interpretability rather than performance is affected. We have now added this to the main text lines 515-518 as well as in Appendix Section J.
>
> | Method | Fusion | Achilles | GPT | Gene2vec |
> |-------------|------------------|------------------|------------------|------------------|
> | EI | 0.093 (0.001) | 0.072 (0.001) | 0.077 (0.004) | 0.071 (0.006) |
> | BioEI-GO | 0.098 (0.000) | 0.085 (0.000) | 0.095 (0.005) | 0.079 (0.004) |
> | BioEI-ONC | 0.091 (0.001) | 0.074 (0.002) | 0.077 (0.008) | 0.073 (0.006) |

---

> ### Author Response · Authors · 2025-11-19
> **Initial Point by Point Rebuttal (and invitation for further discussion)- Response 3 of 3**
>
> **2. Limited Novelty in BO theory:**  We thank the reviewer for highlighting in Strengths section the novelty of our multimodal surrogate modeling. At the same time, we also appreciate the comment in the Weaknesses section that BO theory described in our work is similar to prior works. We would like to clarify that our work introduces two methodological contributions for perturbation design using BO, which to the best of our knowledge, have not been explored previously.
>
> *(a) Multimodal BO for perturbation design:* Prior BO work for perturbation design ([4], [5]) relies on single-modality gene embeddings. In contrast, our approach integrates heterogeneous biological modalities (sequence, network, and functional attributes) and provides analysis showing that this multimodal structure significantly alters the surrogate model’s behavior near local optima. This insight sheds light on why multimodal BO is more efficient—an explanation not exposed in earlier work. Following feedback from other reviewers, we additionally developed more sophisticated latent-space multimodal fusion strategies, which further improve performance; these results are now provided in Appendix F and referenced in lines 192-193 in main text
>
> *(b) Incorporating biological priors into BO acquisition via enrichment analysis:* While enrichment analysis (EA) is widely used biologically, it is inherently purely exploitative. We introduce the first acquisition strategy that integrates EA-derived priors into the π-BO framework while retaining a principled exploration–exploitation balance. This creates a new mechanism for injecting domain-structured priors directly into BO acquisition.
>
> Together, these contributions go beyond a straightforward application of existing techniques: they show how prior can be developed using EA and can be incorporated into BO in a theoretically motivated and empirically validated way while also providing analysis on surrogate model’s behavior near local optima to explain the gains from BO .
>
> Although our work is motivated by biological perturbation design, the methodological developments above are general and applicable to a broad class of BO problems involving structured priors and multimodal embeddings. The ICLR 2026 Call for Papers for the Applications track explicitly lists applications in biology as a core area (https://iclr.cc/Conferences/2026/CallForPapers), and recent BO-for-perturbation-design papers have appeared at ICLR/ICML ([4], [5], [6])—reflecting the growing ML interest in this problem. Thus, while the application area is biological, it is within scope of ICLR 2026 call for papers and the contributions target the broader ML community working on BO, structured priors, and domain-informed BO.
>
> We sincerely appreciate your thoughtful feedback. We have worked to address each point in detail, and we hope the changes meet your expectations. If you agree that the concerns have been resolved, we would greatly appreciate your consideration in raising the score!
>
>
> [1] Destefanis et al., Myc as a Regulator of Ribosome Biogenesis and Cell Competition: A Link to Cancer. Int J Mol Sci 2020. \
> [2] Akef et al., Ribosome biogenesis is a downstream effector of the oncogenic U2AF1-S34F mutation. PLOS Biology 2020.  \
> [3] Onomoto et al., Regulation of RIG-I-like receptor-mediated signaling: interaction between host and viral factors. Cellular & Molecular Immunology 2021. \
> [4] Mehrjou et al.. Genedisco: A Benchmark for Experimental Design in Drug Discovery. ICLR 2021. \
> [5] Lyle et al., Discobax: Discovery of Optimal Intervention Sets in Genomic Experiment Design. ICML 2023. \
> [6] Roohani et al., BioDiscoveryAgent: An AI Agent for Designing Genetic Perturbation Experiments.  ICLR 2025.

---

> > ### Comment · Reviewer_cXha · 2025-11-23
> > **Post-Rebuttal Comments**
> >
> > Thank you for the detailed rebuttal. The clarifications and additional experiments, particularly regarding the fusion strategy and the use of multiple pathway sources, addressed my main concerns. The discussion on pathway dependence and the scalability/runtime analysis was also helpful in clarifying the method’s behavior. Overall, your responses strengthened the paper, and I have updated my score accordingly.

---

> > > ### Author Response · Authors · 2025-11-25
> > > **Thank you for re-evaluation**
> > >
> > > Thank you very much for your thoughtful follow-up and for taking the time to reconsider your evaluation. We genuinely appreciate your constructive feedback throughout the review process, which significantly helped us strengthen the manuscript. We are glad that the additional analyses addressed your concerns and clarified the method's behavior.

---

### Official Review · Reviewer_tp66 · 2025-10-30

**Soundness:** 3
**Presentation:** 3
**Contribution:** 3
**Rating:** 6
**Confidence:** 4

**Summary:**

This paper proposes Biology-Informed Bayesian Optimization (BioBO) for designing genomic perturbation experiments. The method makes two main contributions: (1) integrating multimodal gene embeddings (Achilles, Gene2Vec, GenePT) to improve surrogate modeling, and (2) augmenting acquisition functions with biological priors from enrichment analysis using the πBO framework. Experiments on GeneDisco benchmarks show 25-40% improvement in labeling efficiency compared to standard BO, with the method identifying biologically coherent pathways. The authors demonstrate that multimodal embeddings improve surrogate model performance particularly near the optimum, and provide theoretical no-harm guarantees for their enrichment-augmented acquisition strategy.

**Strengths:**

1.	The combination of multimodal gene embeddings with enrichment-based priors in BO is novel for this application domain. The analysis showing that surrogate model performance near the optimum (rather than globally) correlates with BO performance provides useful insights.
2.	The experimental evaluation is thorough, covering five datasets across multiple modalities and acquisition functions with proper error estimation. The theoretical grounding via πBO provides principled guarantees. The correlation analysis between surrogate quality and BO performance is well-executed.
3.	The paper clearly motivates the problem and explains why existing BO methods for gene perturbation are limited. The two-pronged approach (better embeddings + biological priors) is logical and well-presented.
4.	The problem of efficient gene perturbation design is important for drug discovery. The 25-40% improvement in labeling efficiency could translate to meaningful cost savings. The interpretability aspect through enrichment analysis is valuable for practitioners.

**Weaknesses:**

1.	Limited Technical Novelty: The core techniques are straightforward applications of existing methods. Simple concatenation for multimodal fusion is the most basic approach; more sophisticated fusion architectures (attention mechanisms, learned fusion) are not explored. The enrichment prior construction (Eq. 6) is relatively simple. The paper primarily validates that combining these existing techniques works for this domain rather than introducing new methodology.
2.	Insufficient Biological Validation: Only one detailed biological case study (IFN-γ, Table 2) is provided. The biological interpretations feel somewhat post-hoc rather than predictive. The paper would benefit from: (1) validation with domain experts, (2) wet-lab validation of predictions, (3) deeper mechanistic explanations beyond pathway enrichment, and (4) analysis of failure cases where the method selects biologically implausible genes.
3.	Experimental Limitations: The experiments are restricted to GeneDisco datasets which represent a limited evaluation scope. Key concerns include: (1) DiscoBAX performing poorly contradicts Lyle et al. 2023 without adequate explanation, (2) high variance in some results suggests instability, (3) no comparison with other multimodal fusion approaches or alternative biology-informed methods beyond pure EA, (4) computational costs are not reported, (5) the choice of hyperparameters (beta, temperature, top-10%) lacks principled justification.
4.	Analysis Depth: The correlation analysis reveals interesting patterns but lacks deeper explanation. Why does global LL negatively correlate with BO performance? Why does fusion specifically help near the optimum? The paper states these observations but does not provide mechanistic understanding. The no-harm guarantee is asymptotic, but finite-sample behavior is not analyzed. The sensitivity to beta (Appendix C) shows significant variation that undermines the robustness claims.
5.	Scalability and Generalization: All experiments use the same 10,556 genes common across all embeddings, which is a restricted setting. How does the method perform with missing modalities? What about other organisms beyond human? The batch size is always 1, but practical screens often label batches; how does this affect the approach? The paper does not address these practical considerations.
6.	Presentation Issues: While generally clear, some aspects need improvement: Figure 3 and similar correlation plots are central to the argument but the negative global correlation is counterintuitive and poorly explained. Some claims are overstated (e.g., "dramatically stronger enrichment signals" in Section 4.5 when comparing different experimental settings). The main paper could better synthesize the extensive appendix results.

**Questions:**

1.	Can you provide more detailed analysis or ablation studies on why multimodal fusion specifically helps near the optimum but not globally? What properties of the fused representation drive this?
2.	The DiscoBAX performance is substantially worse than reported in Lyle et al. 2023. Can you investigate and explain this discrepancy? Is it due to implementation differences, hyperparameters, or dataset characteristics?
3.	How sensitive are the results to the choice of top-k% for enrichment analysis? You use 10% but provide no justification or sensitivity analysis for this critical hyperparameter.
4.	Can you provide computational cost comparisons between BioBO and baseline methods? How does the enrichment analysis overhead scale?
5.	For the biological interpretation (Table 2), can you provide additional case studies beyond IFN-γ? Are there cases where the method fails to identify biologically meaningful pathways?
6.	How does performance change when some modalities are missing? This is a practical scenario but not addressed.
7.	Have you validated any of the predicted gene perturbations experimentally or with domain experts? The biological interpretations would be more convincing with external validation.
8.	Why use simple concatenation for fusion rather than more sophisticated approaches like attention-based fusion or learned gating mechanisms? Have you experimented with alternatives?
9.	The beta parameter shows significant impact on performance (Appendix C). Can you provide principled guidelines for setting beta, or ideally, an adaptive approach?
10.	How does the method perform with batch acquisition (B > 1), which is more common in practical CRISPR screens?

---

> ### Author Response · Authors · 2025-11-19
> **Initial Point by Point Rebuttal (and invitation for further discussion)- Response 1of 5**
>
> We thank the reviewer for their positive assessment of our work and appreciating the novelty of our work for the application domain, our evaluations, presentation and the potential real-world impact of our paper.  We first address all the questions raised by the reviewer point-by-point and then address any remaining comments mentioned in the Weaknesses section. We have also included all these points in our paper and marked them in blue.
>
> **1. Why does multimodal fusion specifically help near the local optimum?** This is not because of any properties of multimodal fusion. In fact, this observation holds for different fusion methods and without any fusion as well (see the newly added explanation in Appendix O and Appendix Tables 10 and 11). This is because in BO, the surrogate model is used primarily to estimate the relative ordering of high-value candidates, and hence to identify the local optimum of the objective function near top-performing genes in our application. Global accuracy has little impact on the acquisition function (as has also been shown in other works such as [1]) . Thus, even if global likelihood does not improve, it can still meaningfully enhance BO performance if it sharpens the model locally. Moreover, if we use BO based acquisition, we have more data around the local maximum, therefore the performance of surrogate model has a much better local LL than global LL. This is also verified by the observation that if we only use random acquisition, the correlation between local LL and performance of BO is less higher than using the BO acquisition (also illustrated in Appendix O and Appendix Tables 10 and 11).
>
> **2. Why DiscoBAX performs poorly unlike in Lyle et al. 2023?**  We use the official codebase and hyperparameters provided in the public DiscoBAX repository by Lyle et al, 2023 ([2]). As noted by one of the DiscoBAX authors in a GitHub issue filed approximately two years ago (https://github.com/amehrjou/DiscoBAX/issues/3), an implementation bug affected their originally reported performance, and correcting the bug results in noticeably lower accuracy on these datasets. Our results are consistent with this documented behavior when running their released GitHub code. We had already noted this in a footnote in the main text, and we now have expanded this footnote on Page 6 for clarity.
>
> **3. Sensitivity of results to the choice of top-k% for enrichment analysis:**  We thank the reviewer for raising this point. We have now evaluated BioBO on the IFN-γ dataset (Achilles features) using different top-k% selections for enrichment (5%–50%). Performance is stable for top-k% between 5% and 20%, with only minor variations across acquisitions. We hypothesize that very high top-k% values (e.g., 30–50%) dilute the enrichment signal because including a larger fraction of predicted genes introduces many less informative or marginally relevant genes. Enrichment analysis then averages over a broader, noisier set, reducing the contrast between top-performing pathways and the background. In contrast, smaller top-k% values (5–20%) focus on the most promising candidates, yielding stronger and more specific pathway signals, which in turn guides the acquisition function more effectively. Based on these results, we recommend top-k% ≈ 10% as a practical default, though moderate deviations remain effective, demonstrating that BioBO is robust to this hyperparameter choice. We have also included it in Appendix H now and referred to it in main text (lines 230-231).
>
> | Top-k% | 5% | 10% | 15% | 20% | 30% | 50% |
> |--------|----|----|----|----|----|----|
> | BioEI-GO | 0.090 ± 0.001 | 0.085 ± 0.006 | 0.084 ± 0.009 | 0.085 ± 0.010 | 0.074 ± 0.002 | 0.071 ± 0.001 |
>
> **4. Computational cost comparisons between BioBO and baseline methods & how does EA increase compute overhead?:** Runtime per iteration of our BioBO  is comparable to existing BO methods (we have now included a runtime table for clarity in Appendix G and referred to it in Section 4.6 in main text). We also report it below. The table includes variants with and without multimodal fusion and enrichment analysis (EA) to show the computational overhead introduced by these two contributions of ours. While multimodal fusion and EA slightly increase runtime compared to single-modality (Achilles) models with different acquisition functions, the additional cost remains modest and practical for typical high-throughput CRISPR experiments.
> | Method | Average Runtime per BO Cycle (s) |
> |--------|---------|
> | UCB (Achilles) | 8.55 |
> | UCB (Fusion) | 10.50 |
> | BioUCB-HM (Fusion) | 12.45 |
> | EI (Achilles) | 7.64 |
> | EI (Fusion) | 12.57 |
> | BioEI-HM (Fusion) | 13.05 |
> | TS (Achilles) | 6.95 |
> | TS (Fusion) | 12.18 |
> | BioTS-HM (Fusion) | 12.87 |
>
> Thus, BioBO is quite fast to run and also identifies high-value perturbations more efficiently.

---

> ### Author Response · Authors · 2025-11-19
> **Initial Point by Point Rebuttal (and invitation for further discussion)- Response 2 of 5**
>
> **5 (a) Additional case studies beyond IFN-γ:** As per reviewer's suggestion, we have now analyzed another immune-cell perturbation dataset  IL-2 (details in Appendix I and referred to in main text in lines 466-469). The conclusions are similar to IFN-γ dataset: baseline UCB identifies some relevant pathways with modest enrichment, while our BioUCB-HM highlights immune-relevant pathways with substantially stronger enrichment signals. On this immune-cell dataset, baseline UCB identifies relevant pathways such as MYC_TARGETS_V1, E2F_TARGETS, G2M_CHECKPOINT, and DNA_REPAIR, with overlaps of 49/200, 36/200, 26/200, and 22/150, and adjusted p-values of 6.16×10⁻²⁷, 8.76×10⁻¹⁵, 2.01×10⁻⁷, and 2.41×10⁻⁷, respectively. In contrast, BioUCB-HM highlights MYC_TARGETS_V1, E2F_TARGETS, MYC_TARGETS_V2, and G2M_CHECKPOINT, with substantially higher overlaps of 179/200, 41/200, 18/58, and 32/200, and stronger significance (adjusted p-values 2.39×10⁻²³¹, 2.45×10⁻¹², 1.79×10⁻⁸, and 5.50×10⁻⁷). These results indicate that BioUCB-HM not only recovers relevant pathways found by UCB but also significantly strengthens the enrichment signal for key regulatory circuits. Mechanistically, the enriched pathways are central to T cell proliferation and effector function: MYC regulates cell growth and metabolic activity [3, 4], while E2F and G2M checkpoint pathways control cell-cycle progression [5, 6]. The stronger enrichment observed with BioUCB-HM reflects its ability to prioritize perturbations that are functionally relevant, providing deeper mechanistic insight into immune-cell regulation.
>
> **5 (b) Failure cases:**  BioBO’s enrichment-driven acquisition depends on the relevance of the pathway database. If the database is mismatched to the experimental context—for example, using oncology-specific pathways to guide immune-cell perturbations—then the enrichment prior may bias selection toward irrelevant genes. In such scenarios, the acquisition effectively reduces to the underlying surrogate model (multimodal embeddings), and pathway-level interpretability is diminished.  When a mismatched resource is used such as oncology-specific pathways (“ONC”) applied to immune-cell experiments, the enrichment prior becomes uninformative. For clarity, we show below that while BioEI-GO (the correct enrichment prior) improves performance consistently, BioEI-ONC (the incorrect enrichment prior) yields little or no gain relative to EI.  We have now added this to the main text lines 515-518 as well as in Appendix Section J.
>
> | Method | Fusion | Achilles | GPT | Gene2vec |
> |--------------|----------------|----------------|----------------|----------------|
> | EI | 0.093 (0.001) | 0.072 (0.001) | 0.077 (0.004) | 0.071 (0.006) |
> | BioEI-GO | 0.098 (0.000) | 0.085 (0.000) | 0.095 (0.005) | 0.079 (0.004) |
> | BioEI-ONC | 0.091 (0.001) | 0.074 (0.002) | 0.077 (0.008) | 0.073 (0.006) |
>
> **6. Performance change when some modalities are missing:**  We thank the reviewer for raising this important practical scenario. We evaluated BioBO’s robustness by simulating missing modalities and imputing missing embeddings KNN. Table below (also included in Appendix K now and referred to in main text in lines 518-520) summarizes performance (mean ± std over 7 seeds) across acquisition functions. As shown, multimodal fusion consistently outperforms single modalities, demonstrating that combining heterogeneous embeddings improves surrogate modeling even in the presence of missing data.
>
> | Modality | EI | UCB | TS |
> |----------|----|-----|----|
> | Fusion (all modalities, 18,344 genes) | 0.046 ± 0.003 | 0.060 ± 0.001 | 0.048 ± 0.002 |
> | Achilles only | 0.040 ± 0.001 | 0.042 ± 0.002 | 0.041 ± 0.001 |
> | GPT only | 0.034 ± 0.002 | 0.050 ± 0.008 | 0.041 ± 0.008 |
> | Gene2Vec only | 0.028 ± 0.003 | 0.035 ± 0.003 | 0.033 ± 0.003 |
>
> **7. Validation with domain experts:** Thank you for this point! We have actually validated the findings (particularly related to the interpretability analysis related to immunology datasets) with two independent domain experts who also pointed us to the cited literature to support the mechanistic analyses presented by BioBO. We have clarified this in Appendix I now.

---

> ### Author Response · Authors · 2025-11-19
> **Initial Point by Point Rebuttal (and invitation for further discussion)- Response 3 of 5**
>
> **8. More sophisticated multimodal fusion strategies beyond concatenation:**  We thank the reviewer for suggesting more advanced fusion approaches. We have now implemented latent-space fusion with the three modalities. Each modality \(x_1, x_2, x_3\) is first projected via modality-specific fully connected layers (with dropout for uncertainty), then fused either via concatenation (called latent concatenation) or cross-attention (called latent attention) in the latent space, followed by a final Bayesian MLP.
>
> We report the results for this for  IFN-γ dataset (mean ± std over 7 seeds) below and add this along with results for IL-2 dataset in Appendix F and referred to in main text in lines 193-194. Latent attention consistently outperforms latent concatenation and the Bayesian MLP concatentaion in input space (original fusion strategy in paper) across acquisition functions for both datasets, particularly for BioUCB-HM. This confirms the benefit of integrating multiple modalities and reinforcing our prior results even further.
> | Model | EI | BioEI-GO | BioEI-HM | TS | BioTS-GO | BioTS-HM | UCB | BioUCB-GO | BioUCB-HM |
> |-------|----|-----------|----------|----|-----------|----------|-----|------------|-----------|
> | Bayesian MLP | 0.093 ± 0.001 | 0.098 ± 0.000 | 0.096 ± 0.001 | 0.083 ± 0.001 | 0.095 ± 0.001 | 0.097 ± 0.001 | 0.100 ± 0.001 | 0.102 ± 0.001 | 0.109 ± 0.001 |
> | Latent Concatenation | 0.102 ± 0.001 | 0.107 ± 0.000 | 0.108 ± 0.001 | 0.099 ± 0.003 | 0.105 ± 0.001 | 0.110 ± 0.003 | 0.102 ± 0.003 | 0.101 ± 0.004 | 0.112 ± 0.002 |
> | Latent Attention | 0.109 ± 0.002 | 0.115 ± 0.004 | 0.116 ± 0.003 | 0.107 ± 0.001 | 0.109 ± 0.002 | 0.124 ± 0.002 | 0.113 ± 0.000 | 0.116 ± 0.003 | 0.127 ± 0.003 |
>
> **9. Sensitivity to beta (Appendix C) shows significant variation:** We thank the reviewer for pointing out the sensitivity of β. While performance varies across extreme β values, we observe that β in the range 1-5 yields best performance across acquisition functions (Appendix C) and has relatively minor variations. This is also expected as β controls the extent to which the enrichment prior influences the acquisition score. Small β hence effectively removes the influence of biological prior thus yielding poorer performance compared to using moderate β in the range 1-5. We have expanded the discussion around this in Appendix C.
>
> For adaptive selection, we already set the exponent on the biological prior to β/N (Equation 4 in main text), allowing BioBO to automatically reduce the influence of the prior as the acquisition set grows. This is consistent with the philosophy of Bayesian updating.
>
> **10. How is the performance with batch size > 1:**  We think this is a misunderstanding. In our experiments, each BO iteration already acquires a batch of 20 genes (B=20), not a single gene, to better reflect practical experimental design as the reviewer mentioned. We have now explicitly highlighted it in Appendix Section B under reproducibility subsection.
>
> **Responses to additional comments in Weaknesses section are provided below:**
>
> **1. Justification for selection of other hyperparameters (beta, temperature):** We thank the reviewer for raising this question. Both the temperature parameter t (used to scale enrichment-derived priors) and β (controlling the strength of the prior within the acquisition) modulate the contribution of biological information relative to the surrogate’s uncertainty. Conceptually, t determines how “sharp” or “flat” the enrichment scores are, and therefore how concentrated the biological prior becomes across candidate genes. As t approaches infinity, enrichment based prior approaches uniform distribution and EA is ignored (exploration only) while if t approaches 0,  enrichment based prior approaches dirac-delta distribution and EA is heavily weighted (exploitation only). β controls the extent to which this prior influences the acquisition score. Thus, for practical purposes, both  β and t serve similar purpose. Hence, we keep t=0.1 fixed while in Appendix C, we show the effect of varying β. We observe that very small β effectively removes the influence of biological prior (approaching exploration), whereas extremely large β over-amplifies enrichment signals (approaching exploitation). In practice, moderate values are stable for β. We have now also added this discussion in Appendix L.

---

> ### Author Response · Authors · 2025-11-19
> **Initial Point by Point Rebuttal (and invitation for further discussion)- Response 4 of 5**
>
> **2. Limited Technical Novelty:**  We first thank the reviewer for highlighting in Strengths section that integrating multimodal embeddings with enrichment-based priors is a novel combination for this domain from application perspective. At the same time, we also appreciate the comment in the Weaknesses section regarding the technical novelty of the individual components themselves. We acknowledge that the individual components that form our approach (BO, enrichment analysis) have been studied before but we would like to clarify that our work introduces two methodological contributions for perturbation design using BO, which to the best of our knowledge, have not been explored previously.
>
> *(a) Multimodal BO for perturbation design:* Prior BO work for perturbation design ([2], [7]) relies on single-modality gene embeddings. In contrast, our approach integrates heterogeneous biological modalities (sequence, network, and functional attributes) and provides analysis showing that this multimodal structure significantly alters the surrogate model’s behavior near local optima. This insight sheds light on why multimodal BO is more efficient—an explanation not exposed in earlier work. Following feedback from other reviewers, we additionally developed more sophisticated latent-space multimodal fusion strategies, which further improve performance; these results are now provided in Appendix F and referenced in lines 192-193 in main text
>
> *(b) Incorporating biological priors into BO acquisition via enrichment analysis:* While enrichment analysis (EA) is widely used biologically, it is inherently purely exploitative. We introduce the first acquisition strategy that integrates EA-derived priors into the π-BO framework while retaining a principled exploration–exploitation balance. This creates a new mechanism for injecting domain-structured priors directly into BO acquisition.
>
> Together, these contributions go beyond a straightforward application of existing techniques: they show how prior can be developed using EA and can be incorporated into BO in a theoretically motivated and empirically validated way while also providing analysis on surrogate model’s behavior near local optima to explain the gains from BO .
>
> Although our work is motivated by biological perturbation design, the methodological developments above are general and applicable to a broad class of BO problems involving structured priors and multimodal embeddings. The ICLR 2026 Call for Papers for the Applications track explicitly lists applications in biology as a core area (https://iclr.cc/Conferences/2026/CallForPapers), and recent BO-for-perturbation-design papers have appeared at ICLR/ICML ([2], [7], [8])—reflecting the growing ML interest in this problem.
>
> **3. Experiments on other organisms beyond human:** We are not aware of non-human datasets at the scale of GeneDisco, so we restricted the analysis to human genes. In case, other organisms datasets become available, we will be happy to run the analysis in future for those.
>
> **4. Presentation issues:** Following reviewer's suggestion, we have reframed sentence in line 449 in main text to say "stronger enrichment signals" instead of "dramatically stronger enrichment" in Section 4.5. Also we have now included references to different parts of Appendix in main text and indicated them in blue for easy referencing.

---

> > ### Author Response · Authors · 2025-11-19
> > **Initial Point by Point Rebuttal (and invitation for further discussion)- Response 5 of 5**
> >
> > **5. Why does global LL negatively correlate with BO performance?**  We measure global LL across all genes and observe a weak positive or sometimes negative correlation with BO performance. This occurs because global LL is dominated by the dense region of low-response genes, whereas BO depends almost entirely on the surrogate’s behavior in a small neighborhood around the maximizer. In our CRISPR datasets, only a small fraction of genes exhibit high IFN-γ/IL-2 response. A surrogate that fits the bulk region extremely well (yielding high global LL) but underestimates the tails can therefore perform worse in BO than one that slightly sacrifices global LL but better resolves the local geometry near the optimum.
> > When we restrict LL to the top-k genes (ranked by ground-truth response), the correlation with BO performance becomes positive and substantially stronger (see table below and newly added Appendix O for results). This confirms that BO is driven by local surrogate quality near the optimum rather than global goodness-of-fit. Accordingly, the negative sign in the global LL correlation should not be over-interpreted: it varies across datasets and acquisition functions, and mainly reflects that global LL is not a reliable proxy for BO-relevant model behavior.
> >
> >  IFN-γ — Conditional Correlation Results:
> > | Fusion | Method | LL Global | LL@10% | LL@5% | LL@1% |
> > |-------------------|--------|-----------|--------|-------|-------|
> > | None | EI | 0.051 | 0.211 | 0.202 | 0.212 |
> > | | TS | 0.038 | 0.038 | 0.059 | 0.153 |
> > | | UCB | -0.065 | 0.285 | 0.315 | 0.356 |
> > | | Random | 0.181 | 0.008 | -0.009| -0.035|
> > | Input concat. | EI | -0.018 | 0.313 | 0.292 | 0.298 |
> > | | TS | 0.096 | 0.163 | 0.169 | 0.198 |
> > | | UCB | -0.054 | 0.362 | 0.357 | 0.355 |
> > | | Random | 0.209 | 0.037 | 0.003 | -0.037|
> > | Latent concat. | EI | 0.005 | 0.246 | 0.244 | 0.282 |
> > | | TS | 0.078 | 0.140 | 0.145 | 0.205 |
> > | | UCB | -0.067 | 0.273 | 0.299 | 0.364 |
> > | | Random | 0.181 | 0.008 | -0.009| -0.035|
> > | Latent attention | EI | -0.021 | 0.216 | 0.219 | 0.262 |
> > | | TS | 0.070 | 0.151 | 0.178 | 0.270 |
> > | | UCB | -0.135 | 0.285 | 0.314 | 0.382 |
> > | | Random | 0.181 | 0.008 | -0.009| -0.035|
> >
> > IL-2 — Conditional Correlation Results
> > | Fusion | Method | LL Global | LL@10% | LL@5% | LL@1% |
> > |-------------------|--------|-----------|--------|-------|-------|
> > | None | EI | -0.152 | 0.455 | 0.471 | 0.479 |
> > | | TS | 0.007 | 0.325 | 0.362 | 0.386 |
> > | | UCB | -0.139 | 0.493 | 0.515 | 0.515 |
> > | | Random | 0.088 | 0.224 | 0.233 | 0.208 |
> > | Input concat. | EI | -0.173 | 0.572 | 0.581 | 0.568 |
> > | | TS | -0.023 | 0.439 | 0.457 | 0.441 |
> > | | UCB | -0.217 | 0.540 | 0.559 | 0.546 |
> > | | Random | 0.088 | 0.274 | 0.278 | 0.248 |
> > | Latent concat. | EI | -0.197 | 0.524 | 0.542 | 0.526 |
> > | | TS | -0.026 | 0.375 | 0.404 | 0.402 |
> > | | UCB | -0.192 | 0.479 | 0.506 | 0.500 |
> > | | Random | 0.088 | 0.224 | 0.233 | 0.208 |
> > | Latent attention | EI | -0.153 | 0.552 | 0.568 | 0.559 |
> > | | TS | -0.058 | 0.419 | 0.449 | 0.451 |
> > | | UCB | -0.171 | 0.514 | 0.532 | 0.530 |
> > | | Random | 0.088 | 0.224 | 0.233 | 0.208 |
> >
> > **5. Presentation issues**: Following reviewer's suggestion, we have reframed sentence in line 449 in main text to say "stronger enrichment signals" instead of "dramatically stronger enrichment" in Section 4.5. Also we have now included references to different parts of Appendix in main text and indicated them in blue for easy referencing.
> >
> > We believe the revisions comprehensively address all of the issues you raised. If you find the responses satisfactory, we kindly ask you to consider raising your score!
> >
> > [1] Foldager, et al.,  On the role of model uncertainties in Bayesian optimisation. UAI 2023. \
> > [2] Lyle et al., Discobax: Discovery of Optimal Intervention Sets in Genomic Experiment Design. ICML 2023 \
> > [3] Rathmell, T Cell Myc-tabolism. Immunity 2011. \
> > [4] Wang et al., The transcription factor Myc controls metabolic reprogramming upon T lymphocyte activation, Immunity 2011. \
> > [5] DeGregori et al., Distinct roles for E2F proteins in cell growth control and apoptosis, PNAS 1997. \
> > [6] Ren et al., E2F integrates cell cycle progression with DNA repair, replication, and G2/M checkpoints, Genes & Development 2001. \
> > [7] Mehrjou et al.. Genedisco: A Benchmark for Experimental Design in Drug Discovery. ICLR 2021. \
> > [8] Roohani et al., BioDiscoveryAgent: An AI Agent for Designing Genetic Perturbation Experiments.  ICLR 2025.

---

> ### Author Response · Authors · 2025-11-28
> **Friendly request for feedback on responses to your questions...**
>
> Dear Reviewer,
>
> We greatly appreciate the time and effort you’ve put into reviewing our submission. We have carefully responded to all your concerns. We would be grateful if you could consider raising your score in light of the new experiments, explanations and revision. Thank you again for your time and thoughtful suggestions!
>
> Best regards,
>
> The Authors

---

### Official Review · Reviewer_FsM3 · 2025-11-03

**Soundness:** 2
**Presentation:** 3
**Contribution:** 2
**Rating:** 6
**Confidence:** 3

**Summary:**

This paper proposes to enhance Bayesian optimization for genomic perturbation experiment design with multi-modal gene embeddings and enrichment analysis, essentially incorporating biological knowledge as prior to Bayesian optimization.

**Strengths:**

- The methodology component of the paper is generally easy to follow and clearly presented. Figure 1 is particularly helpful for a high level perception of the proposed framework.

- Ablation studies are presented in the experiment section, which helps the understanding of each component's contribution.

**Weaknesses:**

- Novelty: I think novelty of the proposed methodology is lacking. This work is more about how to incorporate biological knowledge and analysis into existing methods such as $\pi$BO. I think sometimes combining existing methods could be interesting, but if it's more of a realization of an approach in a specific domain, I think it's less interesting. To be honest I'm a little surprised that this work was submitted to this venue instead of some computational biology journal, since it has a quite heavy dose of biological content and analysis.

- Clarity & practicality: Although the methodology component of the paper is generally easy to follow and clearly presented, I find the biological experiment setup not so easy to follow (as a person who does not have abundant biology knowledge). From Figure 1, it seems that there is an online component of the UCB update (wet lab perturbations) being conducted, but I assume this is not actually carried out in the experiment section? How are the unlabeled perturbations evaluated online given a dataset in the experiment? And is the proposed method with wet lab perturbation actually practical given the significant efficiency concern with 20 cycles?

**Questions:**

Please see above

---

> ### Author Response · Authors · 2025-11-19
> **Initial Point by Point Rebuttal (and invitation for further discussion)- Response 1 of 2**
>
> We thank the reviewer for their positive assessment of our work and appreciating the experiments and clarity of our paper.  We address the comments and questions raised by the reviewer point-by-point. We have also included all these points in our paper and marked them in blue.
>
> **1. Novelty:**  We thank the reviewer for raising the question of novelty. We would like to clarify that our work introduces two methodological contributions for perturbation design using BO, which to the best of our knowledge, have not been explored previously.
>
>  *(a) Multimodal BO for perturbation design:* Prior BO methods for perturbation design ([1], [2]) rely on single-modality gene embeddings. Our approach integrates multiple heterogeneous biological modalities (sequence, network, functional attributes) and shows that this multimodal structure meaningfully alters surrogate behavior near local optima, explaining why multimodal BO is more sample-efficient—an insight not available in earlier work. In response to reviewer feedback, we also developed more advanced latent-space fusion strategies, which further improve performance; these results now appear in Appendix F and are referenced in lines 192–193.
>
> *(b) Incorporating biological priors into BO acquisition via enrichment analysis:* While enrichment analysis (EA) is widely used biologically, it is inherently purely exploitative. We introduce the first acquisition strategy that integrates EA-derived priors into the π-BO framework while retaining a principled exploration–exploitation balance. This creates a new mechanism for injecting domain-structured priors directly into BO acquisition.
>
> Together, these contributions go beyond a straightforward application of existing techniques: they show how prior can be developed using EA and can be incorporated into BO in a theoretically motivated and empirically validated way while also providing analysis on surrogate model’s behavior near local optima to explain the gains from BO . Furthermore, ICLR 2026 Call for Papers for the Applications track explicitly lists applications in biology as a core area (https://iclr.cc/Conferences/2026/CallForPapers), and recent BO for perturbation design papers have appeared at ICLR/ICML ([1], [2], [3])—reflecting the growing ML interest in this problem. Other reviewers (tp66, cXha, ohwq) also highlighted the relevance of this problem in real-world applications. Thus, while the application area is biological, it is within scope of ICLR 2026 call for papers.
>
> **2. How are unlabeled perturbations evaluated online?:** In our evaluation, we do not perform wet-lab perturbations ourselves. Instead, we simulate the online BO loop by querying from a pool of genes with pre-measured experimental phenotypes, following standard practice in BO for perturbation design (e.g., [1, 2, 3]). Although in real-world, BO would operate on unlabeled genes, retrospective evaluation on fully labeled datasets is necessary to quantify performance. Here, “unlabeled” simply refers to genes not yet selected by the BO algorithm. This evaluation protocol matches prior BO work on gene perturbation design ([1–3]) and is standard in BO and active learning more broadly in other domains ([4, 5, 6]). We have added this clarification to the main text (lines 108-113).

---

> ### Author Response · Authors · 2025-11-19
> **Initial Point by Point Rebuttal (and invitation for further discussion)- Response 2 of 2**
>
> **3. Efficiency of 20 cycles with wet-lab perturbation:**  The choice of 20 acquisition cycles follows exactly the experimental protocol established in prior works ([1,2]), ensuring comparability. In practice:
>
>  (a) Runtime per iteration of our BioBO  is suitable for practical purposes (we have now included a runtime table for clarity in Appendix G and referred to it in Section 4.6 in main text). We also report it below. The table includes variants with and without multimodal fusion and enrichment analysis (EA) to show the computational overhead introduced by these two contributions of ours. While multimodal fusion and EA slightly increase runtime compared to single-modality (Achilles) models with different acquisition functions, the additional cost remains modest and practical for typical high-throughput CRISPR experiments.
>
> | Method | Average Runtime per BO Cycle (s) |
> |--------|---------|
> | UCB (Achilles) | 8.55 |
> | UCB (Fusion) | 10.50 |
> | BioUCB-HM (Fusion) | 12.45 |
> | EI (Achilles) | 7.64 |
> | EI (Fusion) | 12.57 |
> | BioEI-HM (Fusion) | 13.05 |
> | TS (Achilles) | 6.95 |
> | TS (Fusion) | 12.18 |
> | BioTS-HM (Fusion) | 12.87 |
>
> (b) Also note that after 20 cycles, we only select a total of 400 gene perturbations which still corresponds to  less than 5% of the typical gene pool, aligning with realistic experimental budgets in high-throughput CRISPR screens [1, 2, 3].
>
> Thus, BioBO is quite fast to run and also identifies high-value perturbations more efficiently.
>
> We hope that our revisions and additional analyses satisfactorily address all of your concerns. If so, we would be grateful if you would consider updating your score accordingly. Thank you very much for your time and thoughtful review!
>
> [1] Mehrjou et al.. Genedisco: A Benchmark for Experimental Design in Drug Discovery. ICLR 2021. \
> [2] Lyle et al., Discobax: Discovery of Optimal Intervention Sets in Genomic Experiment Design. ICML 2023. \
> [3] Roohani et al., BioDiscoveryAgent: An AI Agent for Designing Genetic Perturbation Experiments.  ICLR 2025. \
> [4] Filstroff et al., Targeted Active Learning for Bayesian Decision-Making, TMLR 2024. \
> [5] Tran-The et al., Bayesian Optimistic Optimisation with Exponentially Decaying Regret, ICML 2021. \
> [6] Li et al., Unlabeled data selection for active learning in image classification, Scientific Reports 2024.

---

> ### Author Response · Authors · 2025-11-28
> **Friendly request for feedback on responses to your questions...**
>
> Dear Reviewer,
>
> We greatly appreciate the time and effort you’ve put into reviewing our submission. We have carefully responded to all your concerns. We would be grateful if you could consider raising your score in light of the new experiments, explanations and revision. Thank you again for your time and thoughtful suggestions!
>
> Best regards,
>
> The Authors

---

### Author Response · Authors · 2025-11-30
**Author message for AC summarizing the rebuttal**

Dear AC,

We would first like to thank you for your time and for navigating an especially challenging review process this year given the unique circumstances. We appreciate the opportunity provided by the conference to submit a final message to assist your assessment.

**Paper summary:** We propose `BioBO`, a biology-informed Bayesian optimization framework for efficient genomic perturbation design. `BioBO` integrates multimodal gene embeddings and enrichment analysis to prioritize high-impact gene knockouts in CRISPR experiments. The approach is interpretable, theoretically grounded, and practically effective.

**Initial review and highlighted strengths:** Our submission received largely positive initial reviews: *8 (accept)* from ohwq, *6 (marginal accept)* from tp66 and FsM3, and *4 (marginally below accept)* from cxHa. Reviewers praised:
- Strong biological motivation and practical relevance – *ohwq, tp66, cxHa*
- Principled integration of enrichment analysis into `πBO` with theoretical guarantees – *ohwq, cxHa*
- Multimodal surrogate modeling – *tp66, FsM3, cxHa*
- Biological interpretability and validation of selected genes – *ohwq, cxHa*

**Post-rebuttal reviewer outcomes:**

Reviewer cxHa increased their score from 4 to 6 after reviewing the new experiments and methodological clarifications.
Reviewer ohwq (score 8) confirmed that all concerns were resolved and reaffirmed their positive evaluation.
The other two reviewers did not respond further, but all of their questions were directly addressed in revision and responses and they had positive assessment from initial review.

**Key reviewer concerns/questions and rebuttal resolutions which led to score increase during the discussion :**

### 1. Advanced multimodal fusion strategies for BO (*tp66, ohwq, cxHa*)
Reviewers suggested exploring alternatives to input-space concatenation of modalities. We implemented `Latent Concatenation` and `Latent Attention` (Appendix F), projecting each modality via modality-specific layers with dropout before fusion and Bayesian MLP prediction. `Latent Attention` consistently improves performance across acquisition functions and datasets, confirming the practical benefit of our multimodal design.

### 2. Scalability and runtime (*cxHa, tp66*)
We added a runtime and scalability analysis (Appendix G), showing that `BioBO` scales efficiently to genome-wide experiments. Overhead from multimodal fusion and enrichment priors is modest, with surrogate training scaling with observed labels rather than search space size, and acquisition linear in candidates.

### 3. Robustness to missing modalities (*tp66*)
As suggested by reviewer, we verified our method's effectiveness when some modalities maybe missing. We provided simulations of missing embeddings demonstrating that `BioBO` with multimodal fusion maintains strong performance (Appendix K), confirming robustness and reduced dependence on any single modality.

### 4. More biological case studies and interpretability (*tp66, cxHa*)
We conducted additional case study on IL-2 dataset with comparative enrichment analysis (Appendix I). `BioBO` consistently identifies perturbations with stronger enrichment in biologically relevant pathways (e.g., `MYC` and `E2F` targets), demonstrating both interpretability and mechanistic insight.

### 5. Mechanistic validation and expert review (*tp66, cxHa*)
Independent domain experts validated our mechanistic interpretations, confirming that prioritized genes (e.g., ribosomal proteins, `PCBP2`) form coherent functional modules relevant to the observed phenotypes (Appendix I and N).

### 6. Failure cases  (*tp66*) and enrichment database ensembling (*cxHa*)
We included a failure case analysis (Appendix J) and experiments ensembling multiple pathway databases (`GO` + `Hallmark`, Appendix M).

### 7. Clarifications on Enrichment Analysis integration and sensitivity analysis of hyperparameters (*ohwq, cxHa, tp66*)
We clarified how enrichment analysis guides `BO` toward coherent gene modules, even without external priors, and provided sensitivity analysis as well as recommendations on how to set different hyperparameters controlling prior strength and exploration-exploitation trade-off (Appendix L) in our method.

### 8. Confidence intervals and statistical reporting (*ohwq*)
We now report full 95% CI (Appendix P) rather than ±1 s.e.m. as suggested by reviewer.

Overall, our responses and new analyses addressed all reviewer concerns, strengthened the biological and methodological validation, and reinforced the practical utility of BioBO.


We hope this summary aids your decision. **We kindly ask the AC to consider our thorough revisions and new experiments, which earned praise and score increases from reviewer cxHa and maintained positive assessments from other reviewers, in their final evaluation.** Thank you for your consideration!

Sincerely, \
The Authors

---

### Meta-Review · Area_Chair_YW8S · 2026-01-03

**Summary:**

This paper introduces BioBO, a biology-informed BO for efficient design of gene perturbation experiments.
Reviewers generally agree that the paper is overall well-written, addresses an important domain application, the proposed BioBO scheme is well-motivated, and the advantages of BioBO are reasonably supported by the evaluation experiments.
Furthermore, the utilization of multimodal gene embeddings and the incorporation of enrichment analysis (EA)-based biological prior into the acquisition function are deemed relatively novel and effective.

There have been some concerns regarding the technical novelty of the proposed scheme, as the proposed BioBO mainly draws from and builds on existing techniques.
There were also concerns regarding the lack of discussion on the overall computational cost and the simplicity of the fusion strategy adopted in the current study.
The authors have thoroughly addressed the reviewers' concerns in their point-by-point response, and most major concerns appear to have been sufficiently addressed.

**Reviewer Concerns:**

Some of the major concerns raised by the reviewers include: potentially limited technical novelty, adoption of a simple fusion strategy for the multimodal gene embeddings, lack of computational cost analysis, and the need for further clarification on aspects such as scalability, interpretability, hyperparameter choices, and dependence on predefined pathways, among others.

The authors have thoroughly addressed the reviewers' comments and provided detailed clarifications to their questions and concerns.
Especially, clarifications have been made regarding the concerns regarding limited technical innovation, additional experiments have been performed by incorporating alternative fustion strategies, runtime and scalability analysis have been carried out, and additional case studies have been performed to demonstrate the merits of BioBO.

Based on the authors' rebuttal, most major concerns appear to have been addressed to a reasonable extent.

**Reviewer Scores:**

The reviewer who originally gave the most negative score acknowledged that major concerns have been addressed.
Other reviewers have originally given moderately positive to positive scores, and the reviewer who gave the highest score also acknowledged that all issues have been addressed and the initial misunderstanding has been clarified.
Based on these discussions and the authors' thorough rebuttal, it is likely that the overall ratings might have been further increased if the reviewers had the chance to fully participate in the discussion during the rebuttal & discussion period.

---

### Decision · Program_Chairs · 2026-01-26

Accept (Poster)